# Phylogenomic signatures of repeat-induced point mutations across the fungal kingdom

**Thomas Badet** ⓘ*, **Daniel Croll**\*

Laboratory of Evolutionary Genetics, Institute of Biology, University of Neuchâtel, Neuchâtel, Switzerland

* thomas.badet@unine.ch (TB); daniel.croll@unine.ch (DC)

## Abstract

Fungal genome sizes exhibit more than a 100-fold variation, largely driven by the expansion of repetitive sequences such as transposable elements (TEs). Silencing mechanisms targeting TEs at the epigenetic or transcript level have independently evolved in many lineages. In fungi, repeat-induced point mutation (RIP) targets TEs by recognizing repetitive sequences and inducing mutagenesis. However, the prevalence of RIP across the fungal kingdom and the fidelity of the canonical C-to-T mutation signatures remain unclear. In this study, we address these gaps by tracking shifts in genome architecture across the fungal kingdom. We find that a striking approximately 30-fold increase in genome size within a clade of leotiomycetes is associated with the absence of several RIP-related genes, suggesting a relaxation of genome defense mechanisms during this expansion. To track the impact of genome defenses, we designed a quantitative screen for RIP-like mutation signatures. The phylum of ascomycetes was unique in showing enrichment in mutation signatures in non-coding and repetitive sequences, consistent with a phylogenetically restricted occurrence of RIP-like genome defense systems. Then, we performed a phylogeny-aware association study to identify gene functions associated with RIP-like mutation signatures. We identified a zinc-finger protein as the strongest candidate underpinning a novel mechanism of genome defenses. Our findings reveal the multifaceted drivers of genome defense systems and their close ties to genome size evolution in fungi, particularly in lineages with evidence for recent RIP activity, highlighting how proximate molecular mechanisms can shape genome evolution on deep phylogenetic scales.

## Introduction

Mechanisms underlying variation in genome size across eukaryotes have attracted significant attention ever since the discovery of the C-value paradox and the extent of non-coding DNA in most genomes [1]. With increasing genome size, the non-coding fraction becomes the dominant constituent of the genome [2]. Non-coding DNA is composed in large parts of transposable elements (TEs) capable of transposing or

**Data availability statement:** The metadata and links for the genomes analyzed in the frame of this study are available from the S1 Table. Scripts and datasets are available on Zenodo (https://doi.org/10.5281/zenodo.15425698).

**Funding:** DC was supported by the Swiss National Science Foundation (grants 173265 and 201149 at https://data.snf.ch/grants). The funder had no role in study design, data collection and analysis, decision to publish, or preparation of the manuscript.

**Competing interests:** The authors have declared no competing interest.

**Abbreviations:** CAZymes, carbohydrate-active enzymes; Hp1, heterochromatin protein 1; RIP, repeat-induced point; RLR, repeat-linker-repeat; RNAi, RNA interference; TE, transposable element.

creating additional copies in genomes [3]. TE content variation is the primary source of genome size variation among species [4]. Beyond inflating genome size, active TEs can impair fitness through deleterious insertions and destabilize genome integrity [5,6]. Multiple genome defense systems, acting post-transcriptionally and aiming to control gene expression required for TE mobilization, have independently evolved in different lineages [7,8]. Such genomic defense systems are widespread among eukaryotes and are based on DNA or histone methylation, as well as RNA interference (RNAi) [9–11]. However, to what extent genomic defenses effectively contain TE proliferation remains poorly understood.

Despite active genome defense systems, many fungal lineages experienced genome expansions and TEs recurrently escaped from host genome control [12–14]. Variation in repeat content and TE activity have also been found within individual species highlighting the potential rapid turnovers in defense mechanisms [15,16]. Fungi present a highly heterogeneous clade for mechanisms and effectiveness of genome defenses. Most species in the Saccharomycotina subphylum (including the Baker's yeast *Saccharomyces cerevisiae*) have small and compact genomes nearly devoid of repeats despite lacking DNA methylation and RNAi systems [17–19]. Small genomes have been associated with the colonization of nutrient-rich environments and the capacity to develop as single-celled yeasts [20,21]. However, yeast-forming fungi have emerged multiple times independently across the fungal kingdom and show convergent loss of important metabolic genes and genome streamlining [21–24]. Host-associated fungi also tend to have smaller genomes than their free-living relatives. The genomes of obligate fungal parasites for instance have been shown to harbor small genomes, suggesting that lifestyle might be a major driver of genome evolution [25,26]. Similar observations were made for obligate endosymbiotic bacteria showing reduced effective population size and inefficient selection [27–29]. Therefore, species with small effective population size are predicted to evolve larger genomes as a consequence of ineffective excision of mildly deleterious insertions such as those induced by TE mobilization [27,30]. However, several species including plant pathogenic fungi experienced TE-mediated genome size increases despite the assumption of large effective population sizes [31–33]. Hence, current models fail to predict genome size evolution with TE defenses likely being a poorly modeled factor.

While many fungi have conserved elements of the machinery for DNA methylation, histone modifications, and RNAi-associated pathways, these systems show considerable variation in their composition and completeness across fungal lineages [17,18,34]. These widely shared mechanisms of TE defense are complemented in fungi by specific mechanisms such as targeted mutagenesis of repeated DNA sequences [35]. This process, known as repeat-induced point mutation (RIP), has been first described in *Neurospora crassa* and is responsible for the highest known mutation rate outside viruses [36,37]. In fungal plant pathogens, RIP-mediated mutation rates underpinned rapid evolution of pathogenicity genes located near repeats [38]. RIP activity likely contributed to the emergence of genome compartments with contrasted TE and gene contents [39]. A major cost of the RIP genome defense

mechanism is to hinder the evolution through gene duplication. In the *N. crassa* genome, most paralogs predate the emergence of RIP activity [40]. RIP is thought to be active in the pre-meiotic cell containing both parental nuclei and promotes C→T mutations in targeted repeats in a CpA/TpG dinucleotide context [41,37]. RIP efficiency is influenced by the length and identity of the matched repeat sequences, as well as by the periodicity of interspersed homology between the repeats [42–45]. In *N. crassa*, two mechanistically distinct RIP pathways have been described. One requires the *Rid1*-encoded cytosine methyltransferase and targets repetitive sequences themselves, while a second pathway dependent on the cytosine methyltransferase Dim2 and the histone H3 lysine 9 methyltransferase Dim5, targets the single-copy regions flanked by the repeats. These two pathways likely operate with different dependencies, genomic targets, and potentially distinct biological roles, yet this dual origin and its implications are poorly understood. Noticeably, the genes encoding Dim5, Rid1 and Dim2 have been lost multiple times in the phylum [46–48]. Many genomes of ascomycete fungi show signatures of RIP in repetitive sequences, however what underpins the patchy distribution remains unknown. The trade-off between genome defense benefits and costs for paralog creation are likely creating complex dynamics.

In this study, we tested the hypothesis that RIP activity prevents genome expansion in fungi. For that, we first determined shifts in genome architecture across 1,239 genomes covering the fungal kingdom. We then tested whether gains or losses of the RIP machinery predicts TE proliferation rates and genome size evolution. Finally, we designed a quantitative screen for RIP-associated mutation signatures to predict previously unknown genetic determinants of RIP activity.

## Results

### Genome evolution in the fungal kingdom

To determine the action and signatures of genome defense systems in the fungal kingdom, we sampled a total of 1,239 genomes representing five major phyla and included 16 outgroup species from Oomycota and Stramenopiles (S1 Table). Although distantly related, these lineages were chosen as outgroups to help distinguish fungal-specific patterns from those shared with other eukaryotes. The number of assembled scaffolds ranged from 4 to 83,551 for an average of 1,211 while genome size was on average 35 Mb ranging from 7.3 Mb (*Malassezia restricta*) to 733 Mb (*Gigaspora margarita*) (Figs 1A and S1). We find that BUSCO scores, which estimate genome assembly completeness based on the presence or absence of a set of highly conserved single-copy orthologs, ranged from 46% completeness (*Botryozyma nematodophila*) to 100 (eight Stramenopiles), for an average of 95% across all phyla (S1 Table). Larger genomes contain larger fractions classified as repeats (Fig 1B and 1D) highlighted by the >100 Mb genomes in species from the Mucoromycota subphylum. We identified 21 species for which more than 50% of the assembly is made of repetitive DNA for a genome size ranging from 71,5–773 Mb (top 5.2% larger genomes, S1-S2 Tables). Genomes from the Saccharomycotina subphylum are small and mostly devoid of repeats, with a repetitive fraction accounting for 0%–20% of the genome (average of 2%). We computed for each species the number of sequences annotated as repeats and that share strong identity (>95% sequence identity over intervals ranging from 100 bp to 10 kb, which we term "repeat identity"; see Fig 1C). As expected for species with few or no repeats, 75% of the Saccharomycotina genomes have less than 10 sequences larger than 1 kb sharing >95% identity (265/353 species). In contrast, 83%–98% of the species in the agaricomycetes, dothideomycetes, eurotiomycetes, sordariomycetes or leotiomycetes classes have more than 10 genomic repeats sharing 80%–95% identity over 1 kb in sequence length (S3 Table).

To investigate the evolution of genome defense mechanisms in a phylogenetic context, we sampled 100 BUSCO genes with a species occupancy ≥50% and built a phylogenetic tree for the phylum (Fig 1E, "Methods"). Using phylogenetic independent contrasts, we find that most of the measures of assembly contiguity are highly correlated (Fig 1F). To prevent spurious associations with genome assembly quality, we assessed phylogenetic signals for 12 genome assembly metrics. This step ensures that observed patterns in genome defense features are not confounded by shared evolutionary history or systematic biases in genome quality across related species. We find that all the tested assembly metrics show some

PLOS Biology

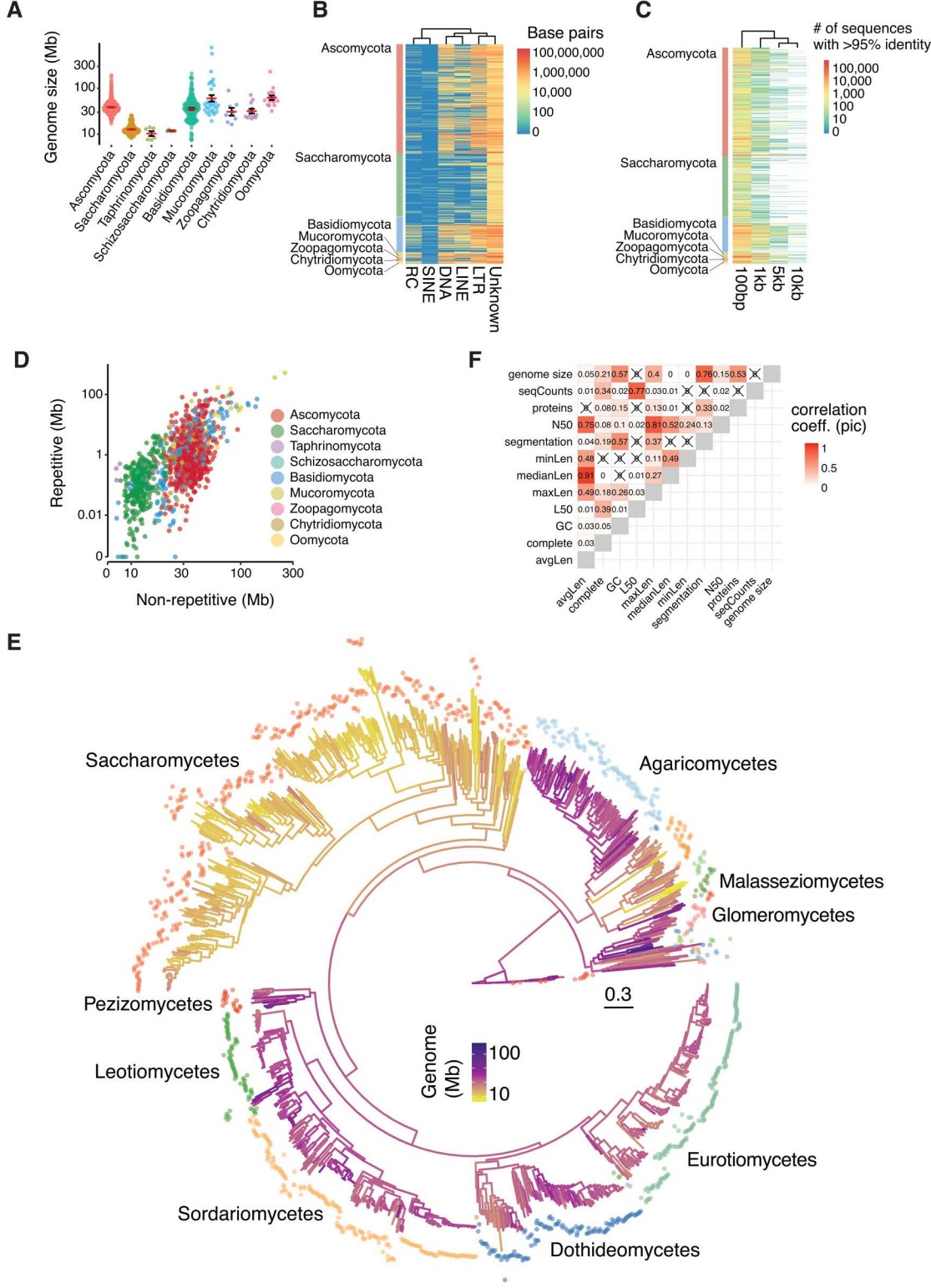

**Fig 1. Genome architecture across 1,239 fungi. (A)** Assembly metrics of the 1,239 genomes. Assembly contiguity varies greatly in the dataset, but except for 13 assemblies from the Oomycota phylum (100 genes in the *Strametopiles* database), all genomes contain more than 100 single-copy BUSCO genes. **(B)** Cumulative length in base-pairs of annotated repeats across the 1,239 genome assemblies. Repeats are classified as Rolling-circle

(RC), Long interspersed nuclear element (LINE), Short interspersed nuclear element (SINE), Long terminal repeats (LTR), DNA transposons (DNA) or unknown. **(C)** Heatmap showing the number of sequences sharing more than 95% sequence identity over 100, 1, 5 and 10 kb across the 1,239 assemblies. Only a small fraction of genomes carries multiple >5 kb repeats with >95% sequence identity. Genomes with >1,000 hits of >95% sequence identity were also among the top 25% largest genomes (31 out of 37 genomes). **(D)** Correlation between repetitive and non-repetitive genome size across the 1,239 assemblies. **(E)** Phylogenetic reconstruction of the relationship based on the concatenated alignment of 100 protein-coding genes. The resulting phylogeny recapitulates all major taxonomic classes represented in the dataset with the Saccharomycetes, Sordariomycetes, Eurotiomycetes, Dothideomycetes and Agaricomycetes classes forming the largest clades. Branch colors represent the maximum likelihood reconstruction of genome size across the phylogenetic tree. **(F)** Correlation of the phylogenetic independent contrasts of different assembly metrics across the 1,239 genome assemblies. We used the reconstructed phylogeny to calculate the phylogenetic independent contrast of each genome assembly metric. Assembly metrics include the average scaffold length (avgLen), percentage of complete BUSCO genes (complete), GC isochore percentage, L50 (smallest number of contigs whose length sum makes up half of genome size), length of the largest scaffold in base-pairs (maxLen), median scaffold length in base-pairs (medianLen), length of the smallest scaffold in base-pairs (minLen), number of GC segments genome-wide (segmentation), N50 (sequence length of the shortest contig at 50% of the total assembly length), number of annotated proteins (proteins), number of scaffolds (seqCounts) and genome size in base-pairs. The data underlying this figure can be found in https://zenodo.org/records/15425698.

level of phylogenetic signal. Applying a local indicator of phylogenetic association, approximately 79% of the species show phylogenetic signal for at least one of the genome assembly or architecture metrics (977/1,239, S2 Fig and S4 Table). However, we find that the phylogenetic signal is evenly distributed along the species tree, suggesting no major taxonomy driven bias in genome assembly quality. We find that genome size strongly correlates with protein numbers and GC dinucleotide content ($r=0.53$ and $0.57$, respectively). We computed genome segmentation metrics based on the number of segments with distinct GC content. We find that genome size and segmentation are highly correlated with larger genomes being more segmented ($r=0.76$, Fig 1F). Using the 3′ and 5′ intergenic distance as a measure of gene density, we show that approximately 12.5% of the genomes show compartmentalization as defined by stretches of contrasted GC content (S3 Fig). In addition, we find that the proportion of the total gene pool found in gene-sparse regions (>5 kb at both 3′ and 5′ ends) is strongly positively correlated with genome size and genome segmentation (S4 Fig).

### Shifts in genome size associated with trophic changes

To dissect drivers of genome evolution, we analyzed shifts in genome architecture in a phylogenetic context. We used a model-based approach that accounts for the shared evolutionary history of species and allows inference of rate variation along the tree. We parametrized a Brownian motion model of trait evolution assuming monotonous and heterogeneous evolutionary rates without *a priori* information on the number or position of rate shifts (*i.e.,* Ornstein–Uhlenbeck process). Analyzing five metrics of genome architecture, we identified a total of 229 shifts across 150 edges of the phylogeny. We also analyze "trophism", defined as the functional trophic profile inferred from the overall composition and abundance of carbohydrate-active enzymes (CAZymes) encoded in the genome, which reflects the organism's capacity to degrade different substrates and thus serves as a proxy for ecological trophism (S5 Table). We excluded 11 edges for which we also identified a shift in an assembly metric, leaving 177 shifts in genome architecture across 139 edges of the phylogeny (Fig 2A–2B and S6 Table). In complement, we also compared assembly quality between species associated with a shift in genome size to a similar set of closely related species in the phylogeny. We find no differences in either N50, L50 or the scaffold number between the two sets, ruling out assembly quality as major bias in the identification of genomic shifts (S5 Fig and S7 Table). The highest number of shifts across the phylogeny were associated with genome size and trophism changes, while most intense shifts were found for the number of highly similar repeat sequences (*i.e.,* repeat identity metric) and the genome segmentation metric (Fig 2A–2B). Overall, 40% of the shifts co-occurred with at least one other metric of genome architecture (70/177; Fig 2B and 2D). Shifts in genome size often co-occurred with changes in trophism (Fig 2B and 2D and S6 Table). Nearly 60% of the shifts are in terminal branches of the tree including three co-occurring shifts at the terminal branch of *Aspergillus parasiticus* with an increase in genome size and repeat identity being associated with trophism change (S6 Table). Shifts tend to occur closer to the tree tips than in preceding non-terminal edges (Wilcoxon

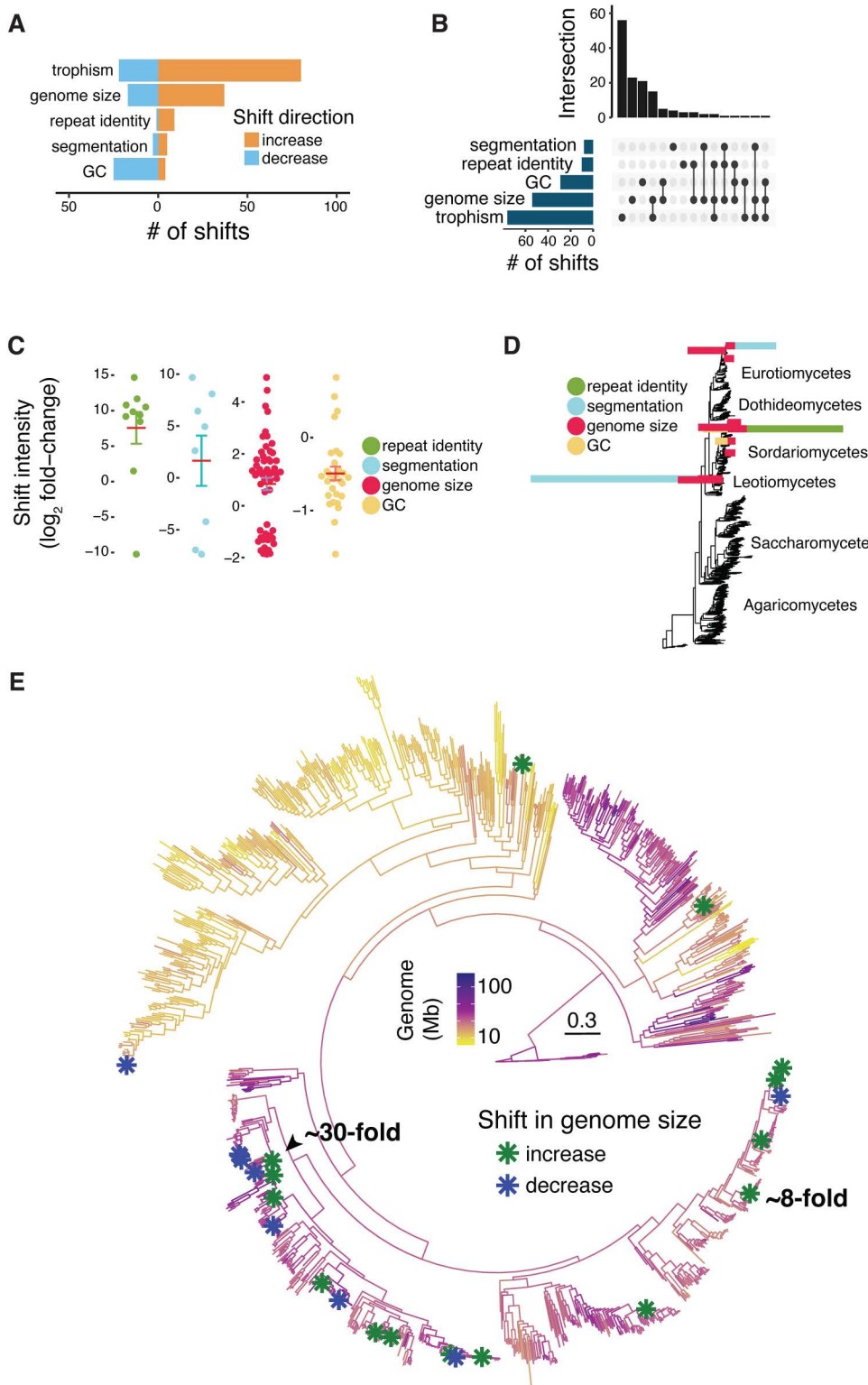

**Fig 2. Widespread shifts in genome architecture and trophism across fungi. (A)** Number of shifts identified across the phylogeny of 1,239 species and for each genomic trait. **(B)** The number of shifts identified for each genomic trait and their co-occurrence. **(C)** Estimated intensity of the shifts identified for each genomic trait. Shifts are expressed as log2 values of the fold-change. **(D)** Visualization of where across the phylogeny shifts co-occur. Bars

are colored by genomic trait variables and sized according to the shifts estimated intensity expressed as log$_2$ fold-change. Fold-change values for shifts in trophism were scaled to 1/1000. **(E)** Phylogenetic positions of the shifts identified for genome size. Negative values denote an identified decrease in the genomic trait value while positive values denote an increase in the genomic trait value. The data underlying this figure can be found in https://zenodo.org/records/15425698.

rank sum test *p*-value = 2.95*e* − 9, S6 Fig). One notable exception was the approximately 30-fold increase in genome size internal to the class of the leotiomycetes with 29 offspring tips (Fig 2E). Following this large increase, five non-terminal offspring branches show shifts in genome architecture, including genome size, repeat identity and GC content. Five shifts in genome size, trophism and repeat identity mapped to a clade of plant pathogens of the *Rhynchosporium* genus suggesting that the gain of the pathogenic trophism might have been facilitated by changes in genome architecture (Fig 3A).

In particular, the 30-fold genome expansion coincides with the inferred loss of the canonical RIP methyltransferase Rid1, for which ancestral state reconstruction suggests that the gene was already absent at the onset of genome size expansion (Figs 3A and S7). We found that both the maintenance-type DNA methyltransferase Dnmt5 and the de novo DNA methyltransferase Dim2 were absent in 13 and 9 species of the same clade of 29 species, respectively (Fig 3A). We confirmed the loss of the RIP-essential cytosine methyltransferase in the enlarged genome of *Blumeria graminis f. sp. tritici* by analyzing the *Rid1* locus of a genome from the sister genus *Cadophora* (Fig 3B). We identified synteny relationships for multiple protein coding genes surrounding the *Rid1* sequence in the *Cadophora* genome matching a single scaffold in the *Blumeria graminis f. sp. tritici* assembly. Five of the 10 genes flanking *Rid1* in *Cadophora* (five upstream and five downstream) were found on the same *B. graminis f. sp. tritici* scaffold. Three of the genes were absent from the assembly, while two were located on other scaffolds. Homology links between translated sequences in the synteny plot between the two scaffolds indicates that, despite local conservation around the *Rid1* locus, *B. graminis f. sp. tritici* clearly lacks the RIP-essential gene (Fig 3B). Across the fungal phylogeny, Rid1 and Dnmt5 are encoded in mostly conserved loci. In contrast, the genomic locus surrounding *Dim2* shows poor conservation across the fungal phylogeny (Fig 3C and S18 Table). This lack of synteny may reflect recurrent gene loss but could also result from translocations or local rearrangements. The disrupted genomic context in multiple lineages suggest that *Dim2* may be under relaxed selective constraint in many fungi. This is consistent with the broader pattern of loss or pseudogenization observed across Ascomycota and supports the idea that Dim2, while involved in multiple cellular functions, may not be universally essential.

## Recurrent lineage-specific losses of genes likely to impact RIP in fungi

In *N. crassa*, RIP is associated with both DNA and histone methylation, though the precise mechanisms remain unclear. We analyzed similar functions across the fungal kingdom to assess potential correlates with genome size shifts. To do this, we constructed a pangenome by clustering all 12,837,225 predicted proteins into orthogroups, which represent sets of homologous genes likely descending from a common ancestor. The resulting 933,315 orthogroups provide a framework to distinguish between core, variable, and species-specific gene content across the dataset (S8–S10 Tables). All species included orthogroups with 10 or more paralogs (S8 Fig and S11 Table). Overall, 79% of the orthogroups were restricted to single species (*i.e.,* unique), and approximately 20% were present in less than 50% of the species (*i.e.,* variable set). Only 2,434 core orthogroups (0.2%) were present in at least 90% of the species making up approximately 12%–68% of each species proteome (Fig 4A and S9 Table). We next investigated the taxonomic distribution of eight orthogroups encoding proteins known to affect RIP in *N. crassa*. The heterochromatin protein 1 (Hp1) and the damage-specific DNA binding protein 1 (Ddb1) are highly conserved in fungi with each being present in >90% of the species (Fig 4B). The histone H3 lysine methyltransferase Dim5 and the ubiquitin ligase components Cullin4 (Cul4) have homologs identified in 72% and 66% of the species, respectively. The two cytosine DNA methyltransferases Rid1 and Dim2 have homologs in only 48% and 51% of the species. Dim2 is nearly absent in the classes saccharomycetes and eurotiomycetes and Rid1 is absent in

**A**

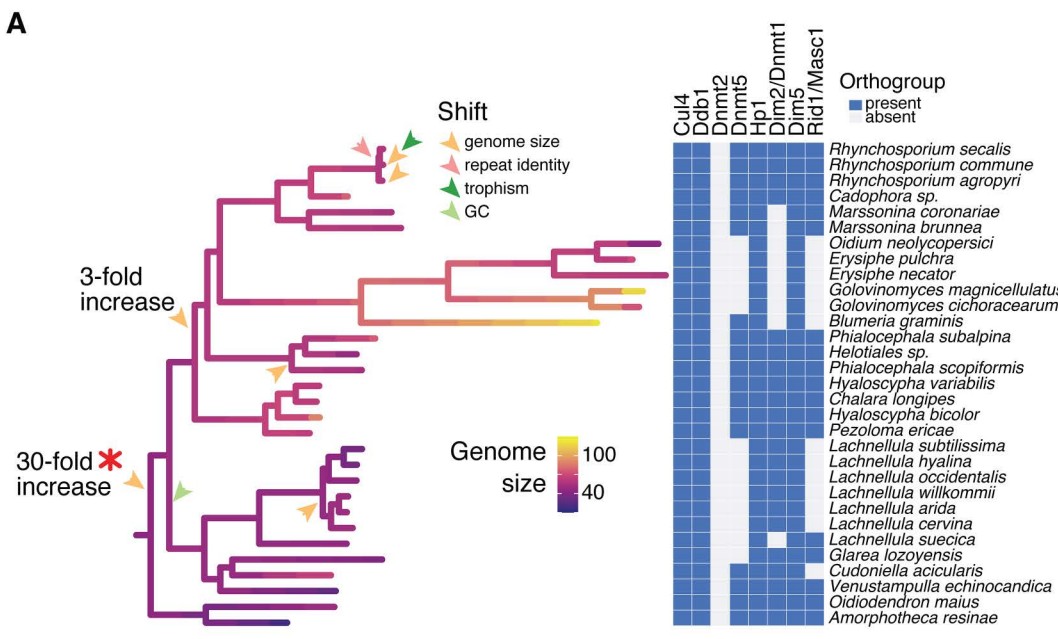

**B**

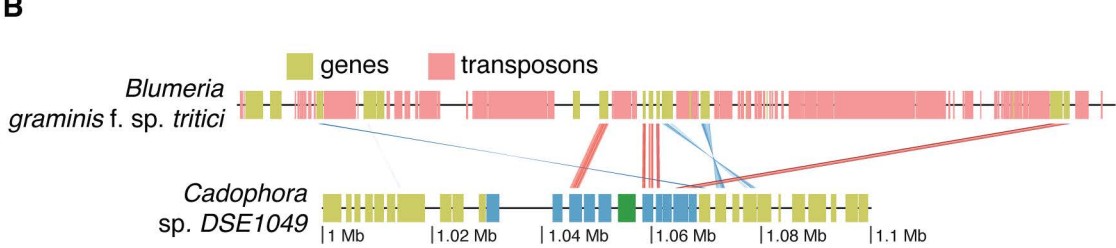

**C**

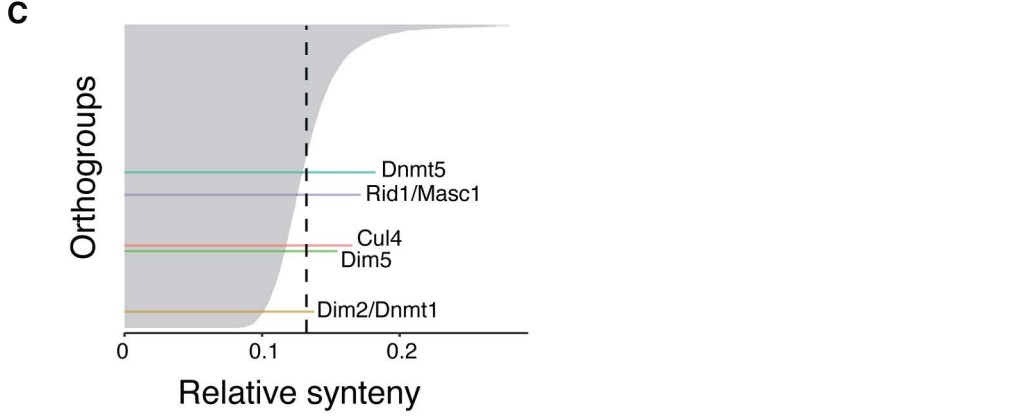

**Fig 3. Major shift in genome size detected in leotiomycetes. (A)** Phylogenetic branch in which the approximately 30-fold increase in genome size occurred in the leotiomycetes. Presence/absence heatmap of RIP-related genes and genes associated with genome size. **(B)** Genomic synteny at the *Rid1* locus between *Blumeria graminis* and *Cadophora* sp. *DSE10*. Syntenic links are reported for encoded amino-acid sequences in all possible reading

frames and are colored according to protein sequence identity computed with *mmseqs easy-search* (--search-type 2 --forward-frames 1,2,3 --reverse-frames 1,2,3; https://github.com/soedinglab/MMseqs2). Red links represent hits with conserved sequence order (same orientation), while blue links indicate inverted (reverse orientation) matches. Darker shades correspond to higher sequence identity between the aligned regions. The *Rid1* gene as annotated in *Cadophora* is shown in green. **(C)** Synteny of 4,666 orthogroups across the 1,239 genomes. The dotted line depicts the average synteny across all orthogroups. The data underlying this figure can be found in https://zenodo.org/records/15425698.

saccharomycetes (Figs 4B and S9). Species carrying Rid1/Masc1 homologs have more segmented and larger genomes, as well as lower repeated sequence identity (Fig 4C). Repeat-targeted mutations have been associated with epigenetic modifications and DNA repair machinery, although the underlying mechanisms vary across fungi. For example, while DNA methylation is involved in some species, others show efficient RIP activity in its absence, and RIP in *N. crassa* operates independently of the canonical recombination machinery [8,43]. For an exhaustive view of relevant molecular machineries present across fungi, we used protein family annotations to identify proteins related to DNA methylation, heterochromatin formation, RNA interference, DNA repair and meiotic recombination (S12 Table). These categories were selected because they represent core components of genome defense systems and are functionally linked to RIP, which occurs during the sexual cycle and involves interactions between chromatin state, DNA damage response, and transcriptional silencing. A cytosine DNA methylase domain is present in 871 genomes and mostly absent from Saccharomycotina subphylum and the Oomycota outgroup (Fig 4D, PF00145.18). Proteins with domains associated with DNA repair were found consistently across most genomes, in concordance with their important role (S10 Fig). Histone-related proteins are highly conserved across fungi, with the exception of the histone-lysine N-methyltransferase E(z), which is missing in species from the eurotiomycetes class (Figs 4D and S10). The C2H2 type master regulator of conidiophore development brlA involved in RNA interference was exclusively found in the eurotiomycetes and sordariomycetes classes. Furthermore, the entire Ascomycota phylum lacks the RNA exonuclease acting as a negative regulator of RNA interference (Fig 4D, ERI1). The pool of protein functions showing overlaps with known components of RIP is taxonomically restricted and shows a patchy distribution.

## Widespread nucleotide composition enrichment in repetitive sequences

Mutagenic mechanisms targeting repeats such as TEs should leave strong signatures in repetitive, non-coding sequences. We designed a screen for mutation signatures against TEs, which is agnostic of the specific sequence patterns generated by the mutagenic mechanism. Short, fixed-length DNA sequences, also called *k-mers*, readily capture genome base composition changes introduced by mutational processes, such as the RIP C-to-T transitions. Under active RIP-like mutagenic process, we expect that specific *k-mers* will be systematically skewed in their frequency in repeated regions relative to other genomic compartments. In contrast, in the absence of directed mutagenesis, the action of neutral drift or random mutation should alter *k-mer* frequencies stochastically, without consistent directional biases towards any genomic compartment. Our approach started by segmenting each genome into coding, non-coding and repeated sequences to assess the abundance of 336 specific *k-mers* (including 16 dimers, 64 trimers and 256 tetramers; Fig 5A). These three genomic compartments are not mutually exclusive as repeated sequences such as TEs can carry genes required for mobilization or only include non-coding sequences. We computed *k-mer* frequencies only for full genome assemblies and after filtering for scaffolds larger than 50 kb to reduce artefacts (S11 Fig). We found that nearly all species have at least one *k-mer* that is >2-fold enriched in non-coding sequences compared to coding sequences (1,027/1,239 species, Fig 5C and S13 Table). A total of 152 tetramers were overrepresented in non-coding sequences with a >2-fold enriched in 1,027 species (S12 Fig). We identified 26 *k-mers* with >10-fold enrichment and these *k-mers* were found across 53 species grouped into seven taxonomic classes (S13–S14 Figs). Finally, we identified a set of 10 *k-mers* >2-fold enriched in non-coding sequences in approximately 40% of all analyzed species. This broadly enriched set of 10 *k-mers*

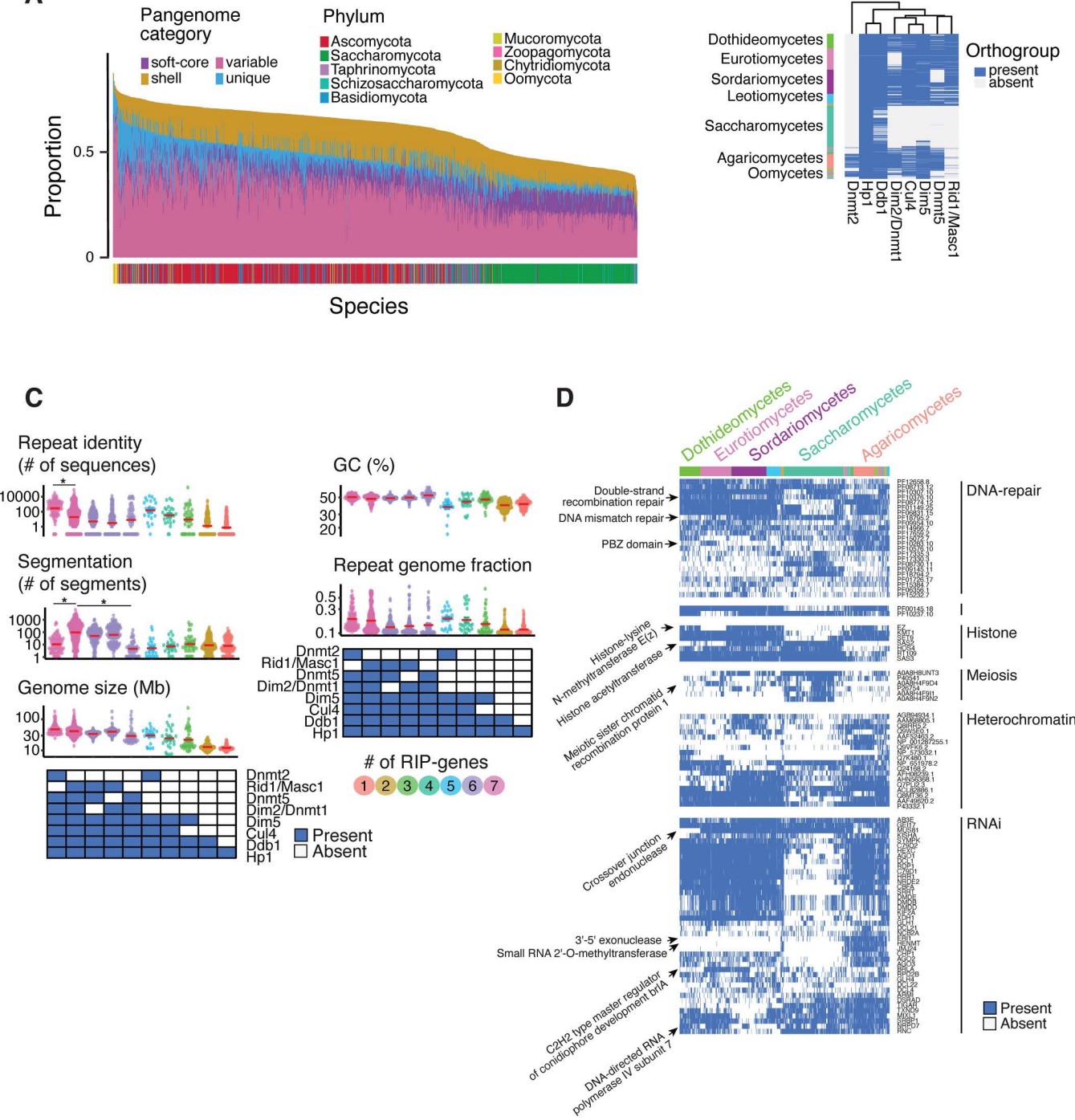

**Fig 4. Patchy distribution of RIP-related protein coding genes across fungi. (A)** Proportion of each of the 1,239 proteomes assigned to pangenome categories. Bars represent, for each species, the proportion of its proteome assigned to pangenome categories. Categories include orthogroups present in 80%–90% of the species (soft-core), 50%–80% (shell), <50% (variable), and orthogroups found in only one species (unique). The core set of proteins (orthogroups present in >90% of species) is not shown. **(B)** Presence/absence heatmap for eight protein coding genes related to RIP in *Neurospora crassa*. **(C)** Variation in six measures of genome architecture in relation to the presence/absence of the eight RIP-related protein coding genes. Presence of the gene is depicted by shaded boxes. **(D)** Presence/absence heatmap for five protein families (Pfams) related to DNA maintenance. The data underlying this figure can be found in https://zenodo.org/records/15425698.

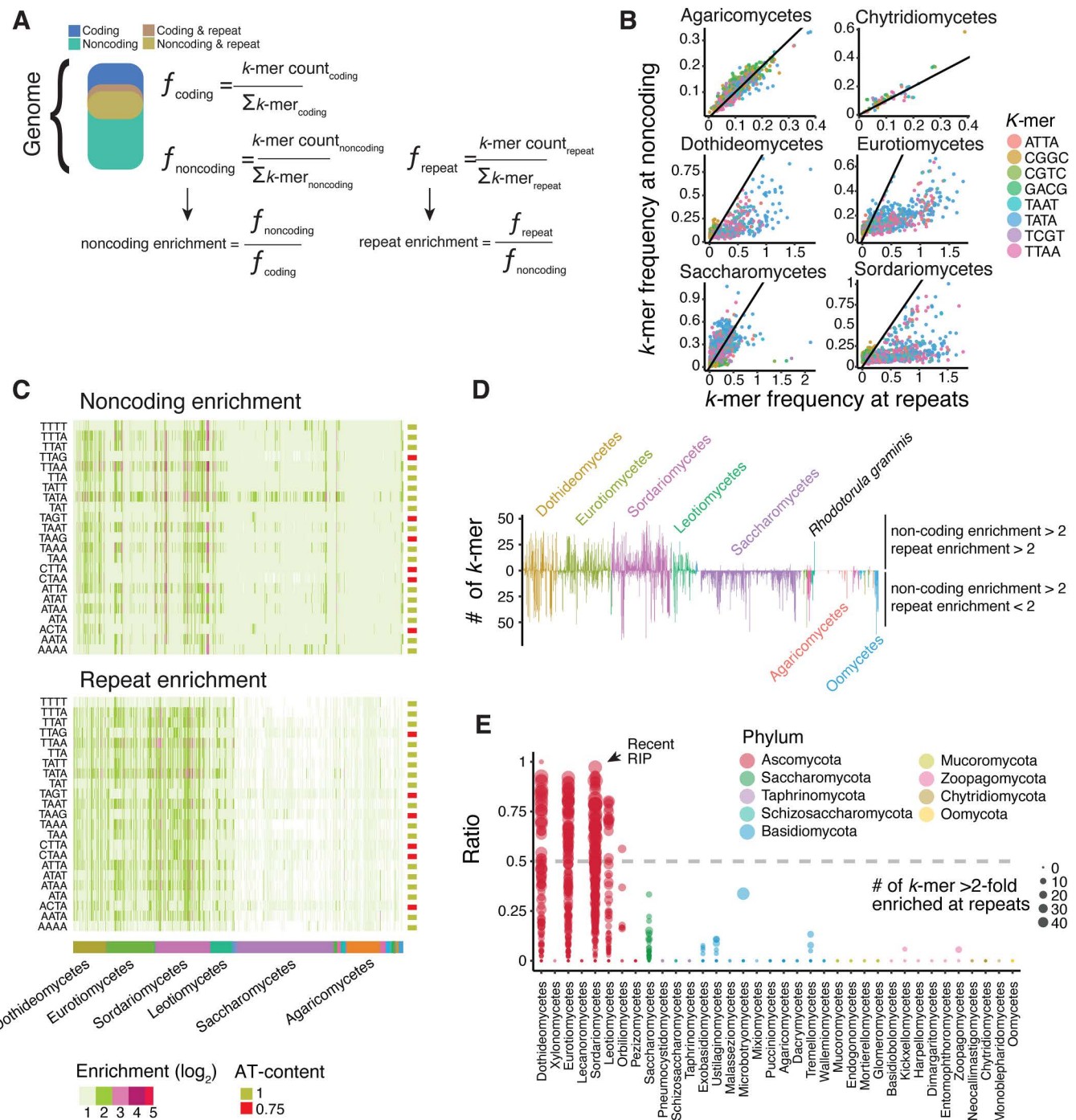

**Fig 5. Identification of repeat-enriched mutation signatures in fungal genomes. (A)** Schematic of the approach used to assess signatures of repeat-targeted mutations. In each of the three non-exclusive genomic compartments, we first counted *k-mers* occurrence that we normalized to frequencies (*f*) by computing the ratio of individual *k-mers* over the sum (Σ) of the *k-mer* count in the compartment. Enrichment values for the non-coding and the repeat compartments were finally computed as frequency ratios **(B)** Correlation between the frequencies at repetitive sequences and non-coding sequences of *k-mers* highly enriched in non-coding sequences. Data is shown for four major taxonomical classes. **(C)** Heatmap for the top 23 *k-mer* enrichments in non-coding sequences compared to coding sequences. **(D)** For each genome, bars show the number of *k-mers* that have both repeat and non-coding enrichment values > 2 (upper bars) or that have only the non-coding enrichment value > 2 (lower panel). **(E)** Proportion of *k-mers* enriched >2-fold in both non-coding and repeat regions across all genomes. Species are grouped by taxonomic class. Values larger than 0.5 is consistent with recent RIP activity. Dot size represents the number of *k-mers* enriched >2-fold at repeats. The data underlying this figure can be found in https://zenodo.org/records/15425698.

were all AT-rich tetramers consistent with sequence signatures generated by RIP mutations (*i.e.,* CpA dinucleotide contexts).

To expand the screen for RIP-like repeat-directed genome defense mechanisms, we analyzed hallmarks of the rapid loss of sequence identity as expected for heavily mutated TEs. For this, we contrasted repeated sequences against the entirety of the non-coding sequences (Fig 5B and S13 Table). We found that in most species the *k-mer* frequency at repeats largely reflects the frequency at non-coding sequences, suggesting that for most genomes the repeated sequences are not targeted by an active RIP-like mechanism (Figs 5B–5C and S15). However, we also identified 68 *k-mers* that are enriched >2-fold in non-coding sequences compared to coding sequences, and simultaneously enriched >2-fold in repetitive regions compared to non-repetitive regions (Fig 5D and S13 Table). In species belonging to the classes dothideomycetes, eurotiomycetes, leotiomycetes and sordariomycetes, AT-rich *k-mers* were found even >5-fold enriched in repeats compared to all non-coding sequences (Figs 5C and S16). In addition, approximately 60% of the species with a strong *k-mer* enrichment in repeats carry only few large sequences with >95% identity (S17 Fig). This matches expectations for a RIP-like mechanism driving the degeneration of repeat sequences (279/466, average number of >5 kb sequences with >95% identity = 23). Interestingly, species with large and repetitive genomes carry on average more repeats of high sequence identity, consistent with recent TE proliferation unopposed by a RIP-like mechanism (S18 Fig).

To formally assess the impact of repeat-directed mutagenesis such as RIP on *k-mer* frequencies, we analyzed experimental crosses in *Neurospora crassa* [49]. In these crosses, parental strains were engineered to contain an 802 bp duplicated sequence separated by a 729 bp linker sequence in the form of a repeat-linker-repeat (RLR) sequence that serves as target for RIP mutations. As controls, we included crosses of strains with deletions for the two key DNA methyltransferases, Rid1 and Dim2 (Δ*rid1*, Δ*dim2*, and Δ*rid1*Δ*dim2* mutants). For each cross, 11–16 progeny were sequenced at the RLR region. Across individual crosses, we detected between 0 and 108 mutations within the RLR sequence. In some progeny, these mutations resulted in up to a 17-fold enrichment of AT-rich *k-mers* (S19 Fig, TAA *k-mer*) specifically within the duplicated sequence, but not in the adjacent linker. Notably, no shift in *k-mer* composition was observed in Δ*rid1* or Δ*rid1*Δ*dim2* crosses, while Δ*dim2* crosses still showed enrichment of AT-rich *k-mers*. These results indicate that Rid1 is required for the RIP-induced skewing of *k-mer* frequencies, and confirm that RIP can leave a strong, localized signature on sequence composition that resemble those identified in repetitive sequences of other fungi.

We assigned each species as putatively RIP-proficient if more than 50% of the *k-mers* enriched >2-fold in non-coding regions were also enriched in repeats (S13–S14 Tables). This threshold reflects the expectation that a mutagenic process like RIP should produce a consistent and widespread signature across both compartments. Importantly, this pattern is also consistent with the known dynamics of RIP, whereby older TE insertions no longer annotated as repeats still carry the mutational signature, leading to similar *k-mer* enrichment in non-coding regions relative to coding sequences. Using a threshold of least 10 *k-mers* with a 2-fold enrichment in non-coding regions and repeats, we found 130 species showing strong signatures of recent episodes of repeat-directed mutations (Fig 5E, 22 dothideomycetes, 45 eurotiomycetes, 50 sordariomycetes and 13 leotiomycetes, respectively). We identified 105 genomes with enriched *k-mers* in non-coding regions and repeats at a ratio of 0.3–0.5. Such genomes experienced most likely only trace activity of a RIP-like mechanism, and the mechanism ceased activity long ago. Using these discrete categories for evidence of RIP-like activity, we found that Ascomycota genomes with trace or ancient RIP activity tend to be smaller than genomes with no traces of RIP (S20 Fig; Tukey's HSD Test for multiple comparisons $p < 9.3e-7$). We also computed pairwise protein sequence identity per genome and found that genomes with trace or ancient RIP activity encoded fewer proteins with recent duplicates (S21 Fig). In conjunction, our results strongly suggest that RIP-like genomic defenses share a preference for CpA dinucleotides and were active in only few fungal clades. Furthermore, RIP-like defenses tend to be associated with smaller genomes and lower protein sequence identity suggesting effective defenses against genome expansion at the cost of constraining gene duplications.

## Phylogenetic associations identify a new candidate gene involved in a RIP-like process

We conducted phylogeny-aware association analyses to identify candidate genes potentially involved in RIP-like mutational processes and broader genome architecture. This method accounts for shared evolutionary histories and controls the risk of false positives due to phylogenetic relatedness. Specifically, we tested associations between orthogroup presence/absence and variation in four genomic features across the phylogeny: genome size, genome segmentation, fungal trophism, and *k-mer* enrichment in repetitive regions (a proxy for RIP-like mutation signatures, S15 Table). Using three independent models, we identified 1,201 orthogroups with significant associations (S16 Table), of which 90 were supported by at least two models and considered robust (Fig 6A). All associations were exclusive to individual genomic characteristics, with fungal trophism accounting for most of the associations (63 orthogroups, Fig 6A). Trophism-associated orthogroups encode lipase, cupin and hydrolase activity matching expectations for their niche adaptation (Fig 6B). Most associated orthogroups reflect the species phylogeny and trophic lifestyles (Figs 6C and S22). Strikingly, many associated orthogroups were missing from the Saccharomycotina subphylum and other monomertrophs in other clades (S22 Fig). Features of genome segmentation associated with 13 orthogroups found in 62–110 species of the Ascomycota phylum, with the majority belonging to the class of sordariomycetes, leotiomycetes and dothideomycetes (136, 50 and 44 species, respectively; Fig 6C). The orthogroups include a total of 1,232 proteins with largely unknown functions with the exception of four orthogroups encoding mostly SKG6, PAN, zinc-finger and CFEM domains (Fig 6B). A further 13 orthogroups are associated with variation in genome size (Fig 6A and 6C). About 20% of all protein sequences included in orthogroups associated with genome size encode AAA+ATPase domains, which are often identified as TE mobilization proteins (S17 Table). This further supports the strong correlation between genome size and TE activity.

Importantly, we identified a single orthogroup (OG0000460) significantly associated with *k-mer* enrichment in repetitive sequences (eta-squared $\eta^2 = 0.4$; Fig 6C–6D), our metric for RIP-like mutational activity. The orthogroup is shared by 598 species ($n = 2,853$ proteins; approximately 97% of Ascomycota excluding Saccharomycotina). The most abundant domain encoded by these proteins is a DNA binding bZIP domain (Fig 6B). Furthermore, genomes carrying the OG0000460 orthogroup show stronger *k-mer* enrichment at repeats as well as larger and more segmented genomes than those without the orthogroup (S23 Fig). In addition, we found that a clade of six species including the plant pathogen *Blumeria graminis f. sp. tritici* with genome size variation of 41–140 Mb lack between 1 and 3 genes associated with variation in genome size (Figs 3A, 6A, and 6C). In *N. crassa*, OG0000460 includes two highly conserved proteins of 260 and 379 amino acids, respectively (Fig 6E). The best structural match for the two *N. crassa* proteins were found with A0A2J8CDV3 (5 members), which includes five members from the fungal kingdom (A0A1V1TG90) in addition to a protein encoded by the plant *Digitaria exilis* (*Poaceae* family). Our results show that no single gene likely controls genome size evolution in fungi. Furthermore, we find that ascomycetes harbor numerous clades with strong evidence for yet undescribed RIP mechanisms underpinning convergent evolution in genome defenses.

## Discussion

Eukaryotic genome size variation is most likely driven by TE activity [50–52]. Here, we showed that genome size evolution can be associated both with the presence of specific orthogroups and sequence signatures of recent RIP activity. Most fungal genomes showed strong *k-mer* compositional enrichment at non-coding sequences compared to coding sequences. However, only a third of the genomes showed a concomitant compositional enrichment both in non-coding and repeated sequences, suggesting that mechanisms of repeat-driven mutations are not conserved across fungi. In ascomycetes, RIP targets repeated sequences to promote C-to-T transitions [42,53]. Consistent with RIP, we found that all repeat-enriched *k-mer* are AT-rich and reminiscent of RIP signatures. Also, we found that species with repeat-enriched *k-mers* were almost exclusively ascomycetes, and this phylum also encodes the clearest molecular toolset for RIP. We used our quantitative measure of mutation bias to pinpoint a gene with a strong positive association to AT-rich *k-mers*. Finally, we observed that an approximately 30-fold increase in genome size in leotiomycetes class, dated to approximately

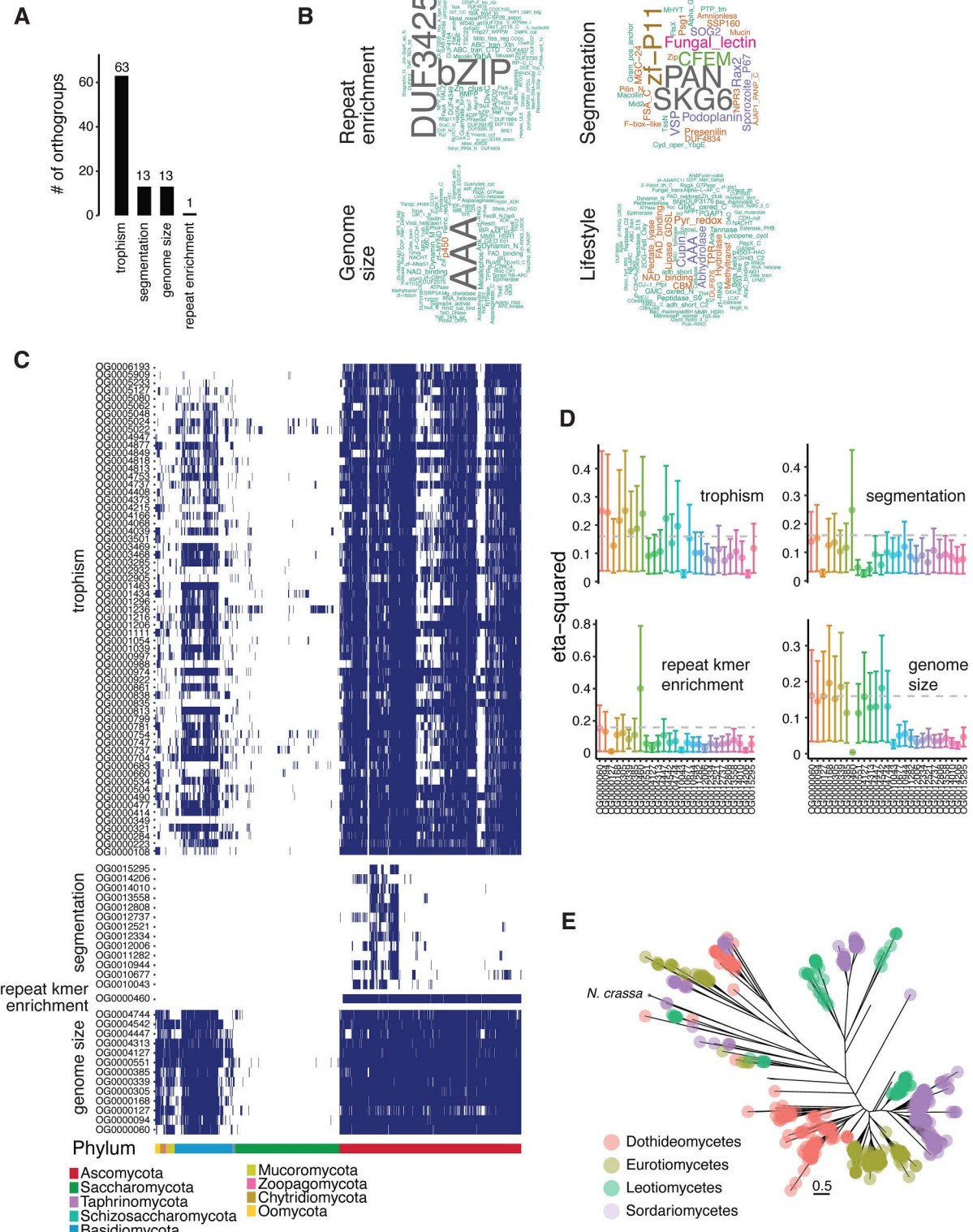

**Fig 6. Identification of protein coding genes associated with repeat-enriched *k-mer* frequencies. (A)** Number of orthogroups associated with genome architecture. **(B)** Word cloud of the most represented protein family annotations of orthogroups associated with genome architecture. **(C)** Presence/absence heatmap of the orthogroups associated with genome architecture. **(D)** Effect size of the top associated orthogroups (presence/absence

variation) with six genomic traits expressed by eta-squared values. **(E)** Phylogenetic relationship of proteins assigned to the orthogroup OG0000460 associated with *k-mer* enrichment at repeats. The data underlying this figure can be found in https://zenodo.org/records/15425698.

120 million years before present, coincided with the loss of several RIP-related genes, suggesting that relaxation of repeat-induced mutational constraints on TEs may have contributed to, or enabled, their accumulation and further genome expansion in this lineage.

Bursts in TEs are often attributed to the inactivation of genomic defenses and the de-repression of active elements [54–56]. Here, we show that RIP activity as observed in *N. crassa* is likely restricted to a small number of ascomycetes. Furthermore, we show that *N. crassa* carries sequence signatures consistent with relict RIP activity, because signatures mostly mapped to non-coding sequences excluding repetitive DNA. Such a mutation bias is most likely explained by weak or dormant TE activity in the species. Using these sequence signatures, we show that most species previously reported to harbor RIP-like mutations do not show biased nucleotide compositions in repeats. For instance, we found no evidence for recent RIP activity in repeats of the TE-rich genome of the barley pathogen *Pyrenophora teres f. teres* with *ca.* 30% of the genome predicted to be RIP-affected [57]. Furthermore, the *Pyricularia grisea* lineage carries signatures of ancient RIP activity but no such evidence in the closely related *P. oryzae*. Even though sexual recombination and RIP-like mutations were documented in *P. oryzae*, most recent divergence occurred through clonal reproduction, which is expected to disable RIP due to the lack of meiosis. This is supported by a recent study showing that recombining and non-recombining populations of the blast fungus *P. oryzae* differ in RIP-like mutation accumulation [58,59]. The *P. oryzae* example illustrates the limitations of using mutational signatures within single genomes to infer the occurrence of RIP in a species.

The paradigm for the genes required for RIP is confirmed in seven species, including *N. crassa*, supporting the existence of a universal toolbox underpinning the genome defense mechanism [58,60–64]. We find however that near 50% of the species lack one of the two important DNA methyltransferases. The cytosine methyltransferase Dim2 and adenine methylation were lost in most eurotiomycetes class. Another cytosine methyltransferase, Dnmt5, was lost in most species belonging to the class sordariomycetes, including *N. crassa* [17,47]. Among the species with experimental evidence for RIP, two most likely encode Dnmt5 (*i.e., P. anserina* and *L. maculans*) showing both weaker RIP signatures [60,64]. Interestingly, most experimental evidence for RIP strength outside of *N. crassa* indicated less effective RIP mutations, in particular in unlinked repeats [58,60–64]. In *P. anserina,* premeiotic recombination and RIP are much more frequent in a deletion mutants with delayed fruiting-body development, further highlighting the differences in RIP activity among fungi [65]. As an example, the DNA methyltransferase Rid1 is essential for RIP but also for normal sexual development in *P. anserina* but not in *N. crassa* [46,66–69]. This is highlighted by the fact that in *P. anserina* the mild RIP-like mutations observed upon sexual recombination are independent from cytosine methylation of the targeted repeats, in contrast to *N. crassa* [64,68].

Our kingdom-wide screen reveals complex dynamics between genome defenses and genome size in fungi. By analyzing over a thousand fungal genomes, we demonstrate that genome size variation is intricately linked to TE accumulation and genome compartmentalization. However, it should be noted that genome size was inferred from assembly size, which is inherently an imperfect proxy and often underestimates the true genome size. While genes essential for genome defense against TEs are frequently lost across the fungal kingdom, this loss is not consistently associated with major TE-driven genome expansion. Instead, ecological factors appear to be the primary drivers of genome evolution, as most changes in genome size coincide with trophism transitions. Within Ascomycota, ancient signatures of RIP activity correlate with smaller genomes, indicating effective control against genome inflation. This kingdom-wide perspective underscores that while RIP mutational signatures are strongly associated with ascomycetes, RIP presence and effectiveness are highly dynamic within the phylum. This highlights the value of comparative genomic studies in uncovering the interplay between genome defense mechanisms and genome architecture. Integrating intraspecific pangenomes into this approach presents

further opportunities to associate TE dynamics with genome evolution. Evolve-and-resequence experiments may reveal host genome responses to TE activity and short-term effects of selection on genome defenses mechanisms [70,71].

## Methods

### Fungal genomes and annotation procedure

The genomes of 1,342 species were retrieved from two different sources (S1 Table). The yeast genomes and their respective gene annotation were taken from the study https://doi.org/10.1016/j.cell.2018.10.023 (available on figshare at https://figshare.com/articles/dataset/Tempo_and_mode_of_genome_evolution_in_the_budding_yeast_subphylum/5854692). All other published fungal and Oomycota genomes were retrieved from NCBI together with their respective gene annotation. We calculated assembly statistics using the *stat* function from the *pyfastx* tool version 0.7.0 [72]. As an additional genome metric, we computed variation in GC content across the assemblies using the *GC-Profile* method [73]. Using a segmentation threshold of 100 and a minimum length of 200 bp, we estimated genome segmentation by counting the number of segments identified by GC-Profile.

### Inference of species trophisms

We categorized species according to their trophic mode as developed by Hane and colleagues [74]. Using gene annotation of the 1,239 individual species genomes, we searched for carbohydrate-degrading enzymes (CAZymes; dbCAN version 10 [75]). We used the *hmmscan* function from the HMMER package version 3.3.2 to identify CAZYmes in each of the proteomes using the dbCAN hidden Markov models as queries [76]. We then applied the *CATAStrophy* algorithm to classify each species trophism given its CAZymes pool. For each proteome we also extract the fitted CATAStrophy principal component values based on its CAZymes frequencies. The first two axes of the principal component analysis were used as a quantitative measure of trophic lifestyles.

### Species phylogeny reconstruction

For the species tree reconstruction, we followed a similar method as in [77]. Briefly, we first identified a set of single-copy orthologous genes in each of the 1,342 genomes using *BUSCO* version 4.1.4 [78] searching the Fungi or Oomycota orthology database version 10, respectively. The method identified a maximum set of 756 BUSCO genes in the genome of the fungus *Colletotrichum plurivorum*. Using a minimal threshold of 50% gene occupancy (>378 BUSCO genes), we proceeded with 1,263 individual species. BUSCO genes were then translated into protein sequences respecting the species' genetic code (code 12 for Saccharomycotina species except for *Pachysolen tannophilus* for which code 26 was used and code 1 for all other species). A random sample of 100 of the resulting protein sequences were then concatenated using the geneStitcher.py script (https://github.com/ballesterus/Utensils) and aligned using the *mafft* alignment tool (version v7.475, --maxiterate 1,000 –auto parameters) [79]. The resulting alignment was trimmed using trimal v1.4.rev15 with the -gappyout option [80]. We estimated the best molecular evolutionary models for the concatenated 100 protein sequences using *partitionfinder* version 2 [81] with the quick option (-q) while using *RAxML* for the analysis version 8.2.12 [82]. The resulting partitioned model was then applied for phylogenetic inference using iqtree2 version 2.1.2 after 1,000 replicates for ultrafast bootstrap and 2 independent runs (-B 1,000 --runs 2) [83]. Finally, the tree was rooted using *Achlya hypogyna* as the oomycete outgroup species with the *root* function in the ape R package v 5.7−1 with resolve.root=TRUE [84]. We further excluded 24 redundant species (*i.e.,* branch lengths of 0) to work on a final set of 1,239 genomes.

### *K-mer* frequency estimates

We computed the frequencies of all possible 336 *k-mer* with size 2, 3 or 4, across all 1,239 fungal genome assemblies. First, we used the gene annotation available for each genome assembly to define coding and non-coding compartments.

Using the *subtractBed* function from the *bedtools* suite v2.26.0 [85], we extracted coordinates of all the intergenic segments for each individual assembly ("gene" was used as the feature to subtract, leaving introns as part of the coding fraction of the genome). Next, we masked the genome assemblies using either the genic or intergenic coordinates (*maskfasta* function from the *bedtools* suite). We then counted *k-mers* in either the gene-masked and intergenic-masked assemblies using *jellyfish* 2.3.0 [86]. To account for variation in assembly contiguity, we also counted *k-mers* after filtering out the scaffolds smaller than 50 kb in length using the *seqtk seq -L* function, version 1.4-r122 [87]. Finally, to account for the size of the coding and non-coding compartments in each genome, we normalized the frequency of each *k-mer* by the sum of all *k-mer* counts in their respective compartment. For each *k-mer*, its non-coding enrichment was represented by the frequency in non-coding sequences over its frequency in coding sequences (*i.e.,* the *k-mer* is defined as enriched in non-coding sequences with values > 1).

## Repeat identification and annotation

To identify putative repetitive elements, we used *RepeatModeler* v2.0.1 using rmblast v2.10.0 in combination with *LTR_Harvest* and *LTR_retriever* (-engine ncbi and -LTRStruct options) [88–90]. For five small genomes we reduced the sampling size (-genomeSampleSizeMax option) to 8,100,000 or 810,000 (Kazsaulg = 8,100,000, Canathen = 8,100,000, Malrestr = 8,100,000, Psehubei = 8,100,000, Kazaerob = 810,000, respectively). Using the consensus sequences identified in each genome, we next annotated repeats with *RepeatMasker* version 4.1.5 using a cutoff value of 250 and skipping bacterial insertion element (-cutoff 250 and -no_is options) [91]. The resulting repeat annotation was filtered for simple_repeats and low_complexity regions, interrupted repeats were merged using the helper script *parseRM_merge_interrupted.pl* and annotations were finally converted to gff3 format using the *rmOutToGFF3.pl* script (https://github.com/4ureliek/Parsing-RepeatMasker-Outputs). Based on the *RepeatModeler* classification, we assigned each putative repeat to either of five major families, namely DNA, Long Terminal Repeat (LTR), Rolling Circle (RC), Long Interspersed Nuclear Element (LINE), Short Interspersed Nuclear Element (SINE), leaving the rest assigned to unclassified (Unknown). To calculate the *k-mer* frequencies at repeats, we used the repeat annotation of each genome assembly to recover their sequences using the *bedtools getfasta* tool [85]. As for the coding and non-coding compartments, we counted *k-mers* at repeat sequences using jellyfish 2.3.0 and calculated for each *k-mer* its normalized frequency using the sum of all *k-mer* counts at repeats within each genome. For each *k-mer*, the repeat enrichment is represented by the frequency at repetitive sequences over its frequency at non-coding sequences, *i.e.,* the *k-mer* is enriched in repetitive sequences for values > 1.

## Repeat sequence identity

To estimate to which extent each genome contains long stretches of high sequence identity, we applied a blast-based strategy. Using the repeat annotation gff files of each genome, we first merged overlapping repeats with the *bedtools merge* function and extracted the resulting repeated sequences using *bedtools getfasta* [85]. The set of repeats of each genome was next blasted against itself using *blastn* from the BLAST 2.10.0 + suite using a word size of 11 and reporting only the top 10 sequences with hits of ≥80% sequence identity (-max_target_seqs 10 -perc_identity 80 -word_size 11). The resulting blastn results were parsed by removing self-hits (query hit against itself) and keeping only the first hit if for a same target sequence multiple overlapping hits with identical alignment score were identified (*i.e.,* same pident, length, mismatch, gapopen and qstart). The remaining hits were then assigned to the following four categories based on their alignment length and sequence identity score: alignment length >100 bp, >1,000 bp, >5,000 bp or >10,000 bp, with sequence identities between 80% and 95% or >95%.

## Genome compartmentalization

To assess the concordance of genome segmentation metric with the "two-speed" genome concept, we estimated genome compartmentalization using gene density. For this, we calculated for each gene in each genome the distance to the

nearest gene in the 5′ and 3′ context. We defined regions where coding sequences are flanked on both sides by intergenic sequences >5 kb as gene sparse. We considered genomes with >1% of their gene pool with flanking intergenic regions >5 kb as compartmentalized.

## Phylogenetic signal and independent contrasts

We estimated phylogenetic signals for the 12 genome assembly metrics using the *phylosignal* package in R (version 1.3, [92]). We first concatenated phylogenetic relationship and trait variables into a phylo4d object using the *phylo4d* function implemented in the phylobase package and assessed the signal for each trait using the five implemented methods by the *phyloSignal* function, namely, Blomberg's K and K*, Abouheif's Cmean, Moran's I, and Pagel's Lambda (https://github.com/fmichonneau/phylobase). In addition, we computed the correlation between the same 12 genome assembly metrics using the phylogenetically independent contrasts (method by Felsenstein 1985 [93]) and implemented in the *ape* package in R version 5.7−1 [84]. We computed the correlation between each pair of metrics by fitting a linear model between the two independent contrasts (fit = lm(iy ~ ix − 1)) in R. For each linear model, we used the coefficient of determination as a measure of correlation between the two variables (adjusted R-squared), and the *p*-value estimates were adjusted using the false discovery rate method in R.

## Identification of putative shifts in genome architecture

To identify the edges in the phylogeny with putative major shifts in genome architecture, we fitted a modified random walk process of trait evolution modeled by an Ornstein–Uhlenbeck (OU) process and implemented in the *phylolm* package in R version 2.6.2 [94]. For each variable, we fitted the model with $\log_{10}$ transformed values except for null values (kept as 0), as well as the PC1 and PC2 values of the CATAStrophy output. Note that 15 species had no identified repeated sequence, and we used the average value of repeat identity for modelling. For each model, we allowed a maximum of 100 shifts across the phylogeny (nmax = 100 option). Best models were selected based on log-likelihoods using the modified Bayes information criterion (mBIC, [95]). The *OUshifts* function outputs the edge on the tree where the shift occurred and the estimated shifts in the expected value of the trait (pshift and shift values in the output). For each edge with a putative trait shift, we recovered the underlying tips in the tree using the corresponding node with the *offspring* function from the *tidytree* package v0.4.5 [96].

## Gene orthology analysis

We inferred gene orthology across the 1,239 species based on protein sequence identity. For that we used *orthofinder* version 2.4.1, which implements *diamond* for homology search (version 0.9.24) [97]. From the entire set of 13,863,658 single proteins, *orthofinder* retrieved 1,008,244 orthogroups. We assessed pangenome categories based on the frequency of each orthogroup in the dataset. We considered orthogroups present in >90% of the 1,239 species as "core" orthogroups. Orthogroups assigned to the "softcore" category were present in >80% but <90% of the species. The "shell" orthogroups were present in >50% but <80% of the species. Orthogroups present in <50% of the species were considered "variable" and those present in only one species were called "singletons". We found major differences in the orthogroup composition across the different taxonomic groups in the dataset. Saccharomycotes have fewer protein coding sequences than the rest of the dataset but approximately 54% were assigned core proteins. Species in the Oomycota outgroup have a large fraction of their proteome assigned as variable (average of 61% of the proteins). Most orthogroups are well defined in the dataset, as approximately 84% are single-copy and likely representing true orthologs (790,441/933,315). Six orthogroups were present in all 1,239 species of the dataset, and all six are multi-copy and associated with functions related to repeat elements (S8 Table). The number of proteins assigned to an orthogroup ranges from 1 to 43,748 (S11 Table). We identified 11,070 orthogroups with ≥10 representatives in at least one genome, and 142,874 orthogroups with more than one representative in at least one species ("multi-copy", S8 Fig and S11 Table). We identified 15 species for

which >40% of the proteome was constituted of multi-copy orthogroups for genome sizes ranging from approximately 16 Mb to >770 Mb (*Martiniozyma abiesophila* and *Gigaspora margarita,* respectively).

## Annotation of functional domains across proteomes

To identify putative functional domains across the species proteomes, we downloaded the annotated domains hidden Markov models from the Pfam release 31 [98]. We used the *hmmsearch* function from the *Hmmer* package version 3.3.2 to scan all the species proteomes for functional domains (--noali option to speed up the process) [76]. We then filtered the resulting outputs for a minimal bitscore of 50 and a maximal e-value of $1e^{-17}$ using the HmmPy.py script (https://github.com/EnzoAndree/HmmPy).

## Identification of putative determinants of genome architecture

We applied a hypothesis-free approach to identify putative genetic determinants of trait variation in a phylogenetic context. We used the *treeWAS* v1 package in R, a method that performs genome-wide associations between a continuous or binary trait and biallelic genomic loci correcting for the confounding effects of relatedness. We performed association tests for a total of 27 traits, including genomic metric, trophism, and *k-mer* enrichment (S15 Table). For each variable, we performed the association of $\log_{10}$ transformed values except for null values that were set to 0 and the PC1 and PC2 values of the CATAStrophy output, which was kept untransformed. For the genotype matrix, we encoded binary values representing presence/absence of the 10,474 orthogroups identified by *orthofinder* with a minimum frequency of 5% in the dataset (see "Gene orthology analysis"). Similarly, we performed association analyses using a presence/absence matrix of the 18,259 protein families (Pfam) identified using *hmmsearch* with the HMM models of the Pfam release 31 (see "Annotation of functional domains across proteomes"). For each trait, we reported the associated orthogroups and Pfams given the terminal, simultaneous and subsequent association tests performed by *treeWAS* (S16 Table). For the associated orthogroups of interest, protein sequences were aligned using Clustal Omega version 1.2.4 allowing for five iterations (--iterations 5) [99]. Gaps in the resulting alignment were trimmed using *trimAl* v1.4.rev15 and gaps in the resulting alignment were further excluded using the -gappyout option [80]. The phylogenetic relationship among proteins was inferred from the trimmed and filtered alignment using FastTree under the Whelan-And-Goldman 2001 model after 1,000 bootstraps (-boot 1,000 and -wag options) [100]. Based on the annotated HMM domains for each protein assigned to the orthogroup, we also counted the occurrence of Pfams for each protein sequence. For the most co-occurring Pfams, we illustrated the protein topology using a random representative using the *drawProteins* package in R [101].

## Protein family analysis

We focused on a set of protein families related to DNA biology (DNA methylation and DNA repair). For that, we screened the Hidden Markov Models domains identified across the entire set of proteins given a Pfam identifier (see "Annotation of functional domains across proteomes", S12 Table). All proteins with a hit for a given Pfam domain were filtered for non-canonical sequences (*i.e.,* no methionine start, in-frame stop codons, https://github.com/milesroberts-123/extract-weird-proteins). The remaining hits were used to assess presence/absence of DNA-repair and DNA methylation related Pfams in each genome (S12 Table). To estimate presence-absence of the eight *N. crassa* RIP-related genes and their orthogroups, we used the *N. crassa* protein sequences to identify RIP-associated orthogroups. For the list of genes associated with meiotic recombination, heterochromatin, histone biology and RNA interference (S12 Table), we used *diamond* in sensitive mode to identify the best reciprocal blast hit among the full set of 13,863,658 proteins (version 0.9.24). For each of the query proteins, the best reciprocal hit was used to define presence/absence of the corresponding gene for each genome.

## Protein identity analysis

For the 1,239 genomes, we performed *diamond blastp* searches using the set of annotated proteins within a genome as both query and subject (default parameters, version 0.9.24). The resulting alignments were filtered for redundant hits and classified according to their length (≤50, 50 < 100, 100 < 500 or >500 amino acids sequence alignment).

## Orthogroup synteny analyses

We focused on the set of 4,666 orthogroups present in >5% of the dataset, single-copy in >80% of the genomes and represented in at least 10 different taxonomic classes (S18 Table). For each of the 4,666 orthogroups, we recovered the 10 neighboring genes in each direction, the orthogroup assignment and counted their occurrence near the focal orthogroup in the dataset. For each pair of focal orthogroups and nearby orthogroups, we estimated their synteny ratio by dividing the number of species with the orthogroup in range of the focal orthogroup by the total number of species carrying the focal orthogroup. The synteny of each of the 4,666 orthogroups was summarized by the mean synteny of all the orthogroups found in the 10 + 10 neighboring gene range.

## Experimental evidence of RIP-induced shifts in *k-mer* composition

To assess the direct impact of RIP mutations on *k-mer* frequencies, we used the data generated from [49] and available on github (https://github.com/jujushen/NcrassaRID/tree/main). Briefly, the RLR DNA fragment containing an 802 bp repeats with complete or partial homology (R) separated by a 729 bp linker (L) was integrated into the wild-type or the Δ*rid1*, Δ*dim2*, and Δ*rid1*Δ*dim2* deletion strains of each mating-type. Between 10 and 20 single spores resulting from each of the four crosses were isolated and PCR-sequenced for the entire RLR locus. For each of these RLR sequence, we counted *k-mer* occurrence at the linker and duplicated regions separately using jellyfish. We computed the frequency of each *k-mer* by normalizing raw counts by the sum of all counts in the focal sequence. Finally, we report frequency ratios by dividing the *k-mer* frequency in the progeny by its frequency in the parental sequence.

## Supporting information

**S1 Fig. Genome assembly metrics across 1,239 fungi. (A)** Genome assembly size in total base pairs. **(B)** First principal component of the CAZyme repertoire in the genome assembly (trophism PC1) suggests that ascomycetes and basidiomycetes have an extended repertoire compared to other phyla. **(C)** GC content of the genome assembly in percentage. **(D)** Number of annotated protein-coding genes in the assembly. **(E)** Number of segments with contrasted GC dinucleotide content across the assembly (segmentation). **(F)** Second principal component of the CAZyme repertoire in the genome assembly (trophism PC2). **(G)** Average number of introns per gene in each assembly. **(H)** Average intron length (bp) in each assembly. **(I)** Total sequence counts in the assembly (seqCounts). **(J)** Average sequence length in the assembly (avgLen). **(K)** Median sequence length in the assembly (medianLen). **(L)** Count of smallest number of contigs whose length sum makes up half of assembly size (L50). **(M)** Maximum sequence length in the assembly (maxLen). **(N)** Minimum sequence length in the assembly (minLen). **(O)** Sequence length of the shortest contig at 50% of the total assembly length (N50). **(P)** Percentage of BUSCO genes found to be complete in each assembly. Note that Oomycota and Mucoromycota have on average larger genomes and more genes while yeasts in the Saccharomycotina subphylum have smaller genomes and less genes. Intron numbers per gene and GC content greatly varies across the dataset, with yeast species having fewer but longer introns and low GC content compared to other ascomycetes. The data underlying this figure can be found in https://zenodo.org/records/15425698.
(TIFF)

**S2 Fig. Widespread signatures of phylogenetic signal of genome assembly metrics.** Local Indicator of Phylogenetic Association (local Moran's I, $\log_{10}$ transformed for display) for each genome assembly metric as calculated by the

lipaMoran function from the phylosignal R package. Only significant associations are displayed ($p$-value $< 0.05$ based on 1,000 permutations). Phylogenetic signal was inferred using five different statistics, namely, Blomberg's K and K*, Abouheif's Cmean, Moran's I, and Pagel's Lambda). Note that two species in the Mucoromycota with some of the largest genomes in the dataset show strong phylogenetic signal for genome size, L50 and the number of scaffolds (S4 Table, seqCounts). The data underlying this figure can be found in https://zenodo.org/records/15425698.
(TIFF)

**S3 Fig. Estimation of the number of assemblies showing a "two-speed"-like genome architecture.** Percentage of the gene pool per genome that is found in gene-sparse regions as estimated by >5 kb intergenic distances up and downstream of the gene. Genome assemblies with more than 1% of their gene pool ($n = 156$) in such gene-sparse regions are matching a "two-speed" genome architecture. The data underlying this figure can be found in https://zenodo.org/records/15425698.
(TIFF)

**S4 Fig. Genome compartmentalization strongly correlates with genome size.** Correlation of the percentage of the gene-pool (genome) that is found in gene-sparse regions as estimated by more than 5 kb intergenic distances up and downstream of the gene with genome assembly GC content (%), the number of segments with contrasted GC dinucleotide content across the assembly (segmentation), total protein number and genome assembly size (Mb). The data underlying this figure can be found in https://zenodo.org/records/15425698.
(TIFF)

**S5 Fig. No major difference in genome assembly quality between species with an associated shift in genome architecture and their close-relative.** Genome assembly quality given the number of scaffolds, N50 and L50 values for species associated with at a shift in one metric of genome architecture (shift-descendent) compared to close relatives with no shift (closely-related). The data underlying this figure can be found in https://zenodo.org/records/15425698.
(TIFF)

**S6 Fig. Detection of shifts and phylogenetic distances to farthest descendant tip.** The data underlying this figure can be found in https://zenodo.org/records/15425698.
(TIFF)

**S7 Fig. Ancestral state reconstruction of presence/absence of the DNA methylase Rid1 across nodes of the phylogeny.** Ancestral state reconstruction was performed using the ace function (ape package version 5.8 in R), upon the all-rates-different maximum likelihood model of discrete trait. Only nodes with a minimal estimated status probability $> 5\%$ are shown. The data underlying this figure can be found in https://zenodo.org/records/15425698.
(TIFF)

**S8 Fig. Proportion of multi-copy orthogroups identified in each genome.** For each genome, the proportion of orthogroups with counts <10 and >10 protein-coding genes are shown separately.
(TIFF)

**S9 Fig. Phylogenetic relationship of proteins with a DNA methyltransferase domain (PF00145).** Proteins with an HMM hit to the PF00145 domain were aligned with Clustal Omega and gaps removed using trimAl for phylogenetic reconstruction under the Whelan and Goldman model of protein evolution with fastTree. The data underlying this figure can be found in https://zenodo.org/records/15425698.
(TIFF)

**S10 Fig. Presence/absence heatmap of DNA biology related genes.** For each genome, the presence of functional domains (Pfam) or protein-coding genes known to be involved in DNA repair, DNA methylation, histone biology, meiotic

recombination, heterochromatin formation, and RNA interference was assessed by HMM domain or blast search (S10 Table). Most of the species from the Eurotiomycetes and Dothideomycetes lack the SAS2 Histone acetyltransferase, while species from the Sordariomycetes mostly lack the histone deacetylase 3 (Fig 4D, SAS2 and NP_651978.2). Most recombination-related genes are conserved in fungi, with the known exceptions of the Mus81 endonuclease absent from Dothideomycetes and the Hop2-Mnd1 recombination complex that is nearly absent from Sordariomycetes. The replication checkpoint Rad17 and the sporulation protein Spo22 are nearly exclusive to the Saccharomycotina subphylum. We also find that homologs of the meiotic sister chromatid recombination protein 1 (Msc1) are absent from the Oomycota, Chytridiomycota, Mucoromycota and Zoopagomycota. In addition to Msc1, Oomycota also lack Msc7 homologs. Similarly, the DNA helicase Srs-2 has been lost multiple times in fungi, including in the Eurotiomycetes. As opposed to the cytosine methyltransferase protein family, which is mostly conserved in ascomycetes (excluding Saccharomycetes), the adenine methylation domain is conserved in the Saccharomycotina subphylum but nearly absent in species from the Eurotiomycetes class (Fig 4D, PF10237.10). Protein families related to the double-strand recombination repair are mostly missing in species from the Sordariomycetes (Fig 4D, PF10376.10). Similarly, proteins from the DNA-mismatch repair family are mostly conserved in Ascomycota but nearly absent from the Eurotiomycetes (Fig 4D, PF18795.2). Proteins with a PBZ domain, often associated with DNA strand-break repair, are nearly absent from the Ascomycota except for species from the Eurotiomycetes (Fig 4D, PF10283.10). The data underlying this figure can be found in https://zenodo.org/records/15425698.
(TIFF)

**S11 Fig. Impact of assembly scaffold filtering on *k-mer* frequency at coding and non-coding sequences.** Assemblies were filtered for scaffolds larger than 50 kb (*i.e.,* filtered assemblies) and *k-mer* frequency calculated at coding and non-coding sequences (2-, 3- and 4-mers). Although the overall distribution of the frequency values in non-coding sequences is constant across the full and the 50-kb filtered datasets, we find that high frequency *k-mers* tend to be over-represented in coding sequences (1,235 genomes left after filtering due to low assembly contiguity). The data underlying this figure can be found in https://zenodo.org/records/15425698.
(TIFF)

**S12 Fig. Number of >2-fold overrepresented *k-mers* in non-coding sequences compared to coding sequences.** The *y*-axis shows the number of species in which a 2-, 3-, or 4-mers is found >2-fold overrepresented in non-coding sequences. For the 212 species without a single *k-mer* overrepresented in the non-coding compartment, we find at least one representative of all nine phyla or sub-phyla in the dataset. The data underlying this figure can be found in https://zenodo.org/records/15425698.
(TIFF)

**S13 Fig. Intensity of non-coding *k-mer* enrichment across different taxonomic classes.** Proportion of species in each taxonomic class that are enriched at non-coding sequences in the range of 1.3 to 2-fold, 2 to 5-fold, 5 to 10-fold or more than 10-fold. Most taxonomic classes have at least one *k-mer* > 2-fold enriched in non-coding sequences (80% or 29/36), with sordariomycetes, saccharomycetes and dothideomycetes being the most represented classes, in addition to eurotiomycetes, exobasidiomycetes, lecanoromycetes and leotiomycetes. The data underlying this figure can be found in https://zenodo.org/records/15425698.
(TIFF)

**S14 Fig. Taxonomic distribution of highly enriched *k-mers* at non-coding sequences.** Distribution of the species taxonomic classes carrying the respective *k-mer* more than 10-fold enriched in non-coding sequences. The data underlying this figure can be found in https://zenodo.org/records/15425698.
(TIFF)

**S15 Fig. *K-mer* frequency at repeats and non-coding sequences across taxonomic classes.** Frequency of eight highly enriched *k-mer* at repeats versus non-coding sequences. Lines represent the *y*=*x* diagonal. The data underlying this figure can be found in https://zenodo.org/records/15425698.
(TIFF)

**S16 Fig. Number of *k-mers* with repeat enrichment >5-fold across taxonomic classes.** Total number of *k-mers* with repeat frequency >5-fold compared to non-coding sequences for all 36 taxonomic classes. *K-mers* were split according to their AT-content (0, 0.25, 0.33, 0.50, 0.66, 0.75 or 1). The data underlying this figure can be found in https://zenodo.org/records/15425698.
(TIFF)

**S17 Fig. Strong enrichment of *k-mers* at repeats associate with few highly similar repetitive sequences.** Genomes with a value of *k-mer* enrichment at repeats >5-fold as a function of the total number of sequences larger than 5 kb sharing between 95% and 100% sequence identity. Dot colors indicate different taxonomic classes. The data underlying this figure can be found in https://zenodo.org/records/15425698.
(TIFF)

**S18 Fig. High number of highly similar repeats correlates with genome repetitive fraction.** Total number of sequences larger than 5 kb sharing between 80% and 95% or 95% and 100% sequence identity per genome assembly as a function of the fraction of the genome assembly identified as repeats. Dot size is scaled to the size of the non-repetitive genome assembly (in base pairs). The data underlying this figure can be found in https://zenodo.org/records/15425698.
(TIFF)

**S19 Fig. Changes in *k-mer* frequency induced by RIP upon a single cross in *Neurospora crassa*.** Frequency ratios of 12 *k-mers* showing >5-fold enrichment at the RLR sequence in the progeny after crosses with the wild-type parents or the deletion mutants Δ*dim2*, Δ*rid1*, and Δ*rid1*Δ*dim2*. *K-mer* frequencies were calculated at the duplicated (100% sequence identity) and the unique linker region separately. The data underlying this figure can be found in https://zenodo.org/records/15425698.
(TIFF)

**S20 Fig. Genome architecture in species with evidence for recent or old RIP mutational signatures.** The asterisk denotes TukeyHSD post-hoc test significant differences. The data underlying this figure can be found in https://zenodo.org/records/15425698.
(TIFF)

**S21 Fig. Species with no evidence for RIP mutation signatures carry more proteins of high sequence identity.** The x-axis shows protein percentage sequence identity calculated from reciprocal blasts. The y-axis is the scaled density given the number of blast hits. Blast hits with sequence length < 50, between 50 and 100, between 100 and 500 or larger than 500 amino acids are represented in the four facets. The data underlying this figure can be found in https://zenodo.org/records/15425698.
(TIFF)

**S22 Fig. Presence/absence heatmap of orthogroups associated with genome size, *k-mer* repeat enrichment and segmentation.** For each associated orthogroup, presence in the focal genome assembly is represented by filled boxes, with the species corresponding taxonomy and trophism annotated as colored tiles (top and bottom tiles, respectively). The data underlying this figure can be found in https://zenodo.org/records/15425698.
(TIFF)

**S23 Fig.  Genome architecture in species with and without the repeat enrichment-associated orthogroup OG0000460. (A)** TATA *k-mer* enrichment at repeats compared to non-coding sequences in species with or without the OG0000460. **(B)** Proportion of the genome annotated as repeats in species with or without the OG0000460. **(C)** Number of highly similar (>95% sequence identity) repeated sequences larger than 1 kb in species with or without the OG0000460. **(D)** Genome size in species with or without the OG0000460. **(E)** Number of isochore genome segments in species with or without the OG0000460. Asterisks denote significant difference as per a TukeyHSD post-hoc test. The data underlying this figure can be found in https://zenodo.org/records/15425698.
(TIFF)

**S1 Table.  Metadata for all 1,239 genome assemblies analyzed in the study.**
(XLSX)

**S2 Table.  Repeat summaries for all 1,239 genome assemblies analyzed in the study.**
(XLSX)

**S3 Table.  Number of identified sequences sharing 80%–95% or 95%–100% identity across the 1,239 genome assemblies analyzed in the study.** Columns denote the number of across different sequence length (<100, <1, <5 kb or >10 kb).
(XLSX)

**S4 Table.  Local Moran's I metric estimated for the different genome assembly metrics across our 1,239 genomes' phylogeny.** The *p*-values are computed given 1,000 permutations using the lipaMoran function from the phylosignal R package.
(XLSX)

**S5 Table.  Values of trophism representing the CAZyme repertoire across the 1,239 genome assemblies.** For each species, the values corresponding to the first two PCA reductions from the CATAStrophy tools are given.
(XLSX)

**S6 Table.  Positions across the phylogeny (edge_num) where shifts in a genome assembly metric (var column) were identified.** The shift column denotes the direction (sign) and intensity of the shift as calculated by the Oushifts function from the phylolm R package. The *is_tip* column indicates if the shift is located at a terminal edge. The *coshift* column indicates if multiple shifts were mapped at the same edge. The *excluded* column indicates edges for which a shift in one of the assembly metrics was also detected and therefore excluded from subsequent analysis.
(XLSX)

**S7 Table.  Wilcoxon rank sum test *p*-values comparing genome assembly metrics (*variable* column) of species for which a shift was identified to close-relative species with no shift (*shift_edge* column).** L50 (smallest number of contigs whose length sum makes up half of genome size), length of the largest scaffold in base-pairs (maxLen), median scaffold length in base-pairs (medianLen), length of the smallest scaffold in base-pairs (minLen), number of GC segments genome-wide (segmentation), N50 (sequence length of the shortest contig at 50% of the total assembly length), percentage of complete BUSCO genes (complete), number of scaffolds (seqCounts) and genome size in base-pairs (totalBases).
(XLSX)

**S8 Table.  Conserved orthogroups in the 1,239 genome assemblies and their associated protein domains.**
(XLSX)

**S9 Table.  The number of orthogroups assigned to each pangenome category and their frequency across the 1,239 genome assemblies.** Orthogroups present in >90% of the 1,239 genomes were designed as "core" orthogroups. Orthogroups assigned to the "softcore" category were present in >80% but <90% of the species. The "shell" orthogroups

were present in >50% but <80% of the species. Orthogroups present in <50% of the species were considered "variable" and those present in only one species were called "singletons".
(XLSX)

**S10 Table. Summary of the number of proteins assigned per orthogroup.**
(XLSX)

**S11 Table. Number of species with unique or paralog proteins assigned to each orthogroup.**
(XLSX)

**S12 Table. List of genes and Pfams used to estimate presence/absence of DNA biology related function across the 1,239 genome assemblies.**
(XLSX)

**S13 Table. *K-mer* frequency calculated at coding, non-coding and repeated sequences in each assembly filtered for scaffolds larger than 50 kb.** Non-coding enrichment calculated as the ratio of the *k-mer* frequency at non-coding over its frequency at coding sequences. Repeat enrichment calculated as the ratio of the *k-mer* frequency at repeats over its frequency at non-coding sequences. For each *k-mer* we report the proportion of the 1,239 genome assemblies showing noncoding or repeat enrichment larger than 2.
(XLSX)

**S14 Table. Number of *k-mer* > 2-fold enriched at non-coding sequences (*n_noncoding*), repeats (*n_repeat*) or both (*n_both*) across the 1,239 genome assemblies.** The ratio of the number of *k-mer* > 2-fold enriched at both non-coding and repeats over the total number of *k-mer* enriched >2-fold is used to estimate recent repeat-induced point mutation activity (Fig 2E). For each assembly we report the estimated RIP status, *i.e.,* "recent RIP" activity, "old RIP" activity or "no RIP" activity.
(XLSX)

**S15 Table. Metrics of genome architecture and their values used for phylogeny-aware association mapping across the 1,239 genome assemblies.** Metrics include values of non-coding and repeat enrichment of the 8 top-enriched *k-mers*. The genome-wide repeat proportion (*repeat_prop*). The number of >1 kb-length sequences sharing >95% identity (*n_1kb*), N50 (sequence length of the shortest contig at 50% of the total assembly length), average scaffold length in base-pairs (*avgLen*), the total number of proteins (*proteins*), the total number of scaffolds (*seqCounts*), number of GC segments genome-wide (*segmentation*), the genome-wide GC content (*GC*), the PCA reduction of CAZyme content (*trophism_PC1* and *trophism_PC2*) and genome size in base-pairs (*totalBases*).
(XLSX)

**S16 Table. List of orthogroups associated with one of the genome assembly metrics (*variable* column) given the simultaneous, subsequent or terminal model implemented in the treeWAS R package (*mode* column).** The number of models for which we find an association is denote in the *n_association* column (1–3).
(XLSX)

**S17 Table. List of the protein domains most commonly associated (top 1 Pfam) with proteins assigned to orthogroups associated with variation in one of the genome assembly metrics (*variable* column).** The number of proteins with the given Pfam domain is given in the *n_protein* column.
(XLSX)

**S18 Table. Relative synteny for a list of 4,666 orthogroups mostly single-copy across the 1,239 genome assemblies (*i.e.,* in >80% of the species).** Synteny ratios were computed by dividing the number of species with the orthogroup

in range of the focal orthogroup by the total number of species carrying the focal orthogroup. The synteny of each of the 4,666 orthogroups was summarized by the mean synteny of all the orthogroups found in the 10 upstream + 10 downstream neighboring gene range.
(XLSX)

**S1 Data. Source data values for each figure in the manuscript.** Values for each figure are provided as individual excel sheets.
(XLSX)

**S2 Data. Text-based *newick* phylogenetic tree underlying the Fig 1E.**
(NEWICK)

**S3 Data. R-data object underlying the phylogenetic tree in Fig 1E.**
(RDS)

**S4 Data. Text-based *newick* phylogenetic tree of the proteins assigned to the orthogroup OG0000460 associated with *k*-mer enrichment at repeats in the Fig 6E.**
(NEWICK)

**S5 Data. Text-based *newick* phylogenetic tree of proteins with a DNA methyltransferase domain (PF00145) in the S9 Fig.**
(NEWICK)

## Acknowledgments

We thank group members for fruitful discussions.

## Author contributions

TB conceived the study and performed analyses with input from DC; DC provided funding; TB and DC wrote the manuscript. All authors have seen and approved this version of the manuscript.

## Author contributions

**Conceptualization:** Thomas Badet.

**Data curation:** Thomas Badet.

**Formal analysis:** Thomas Badet.

**Funding acquisition:** Daniel Croll.

**Investigation:** Thomas Badet.

**Methodology:** Thomas Badet.

**Project administration:** Daniel Croll.

**Visualization:** Thomas Badet.

**Writing – original draft:** Thomas Badet.

**Writing – review & editing:** Thomas Badet, Daniel Croll.

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
