## [Editor Report · Decision Letter 0]

30 Jul 2025

Dear Dr Badet,

Thank you for submitting your revised manuscript entitled "Phylogenomic signatures of repeat-induced point mutations across the fungal kingdom" for consideration as a Research Article by PLOS Biology.

Your revisions has now been evaluated by the PLOS Biology editorial staff, and I'm writing to let you know that we would like to send your submission out for re-review.

IMPORTANT: When you upload your additional metadata (see next paragraph), please can you attend to the following two issues: a) I note that you have not supplied a marked-up "track changes" version of your manuscript. Please provide this. b) In the rebuttal you repeatedly mention your new analysis of N. crassa as being Fig S18 - I believe that it's actually Fig S19; please correct this.

However, before we can send your manuscript back to the to reviewers, we need you to complete your submission by providing the metadata that is required for full assessment. To this end, please login to Editorial Manager where you will find the paper in the 'Submissions Needing Revisions' folder on your homepage. Please click 'Revise Submission' from the Action Links and complete all additional questions in the submission questionnaire.

Once your full submission is complete, your paper will undergo a series of checks in preparation for re-review. After your manuscript has passed the checks it will be sent out for review. To provide the metadata for your submission, please Login to Editorial Manager (https://www.editorialmanager.com/pbiology) within two working days, i.e. by Aug 01 2025 11:59PM.

Kind regards,

Roli Roberts

Roland Roberts, PhD

Senior Editor

PLOS Biology

rroberts@plos.org

---

## [Decision Letter · Decision Letter 1]

11 Sep 2025

Dear Dr Badet,

Thank you for your patience while we considered your revised manuscript "Phylogenomic signatures of repeat-induced point mutations across the fungal kingdom" for publication as a Research Article at PLOS Biology. This revised version of your manuscript has been evaluated by the PLOS Biology editors, the Academic Editor and two of the original reviewers.

Based on the reviews and on our Academic Editor's assessment of your revision, we are likely to accept this manuscript for publication, provided you satisfactorily address the remaining points raised by the reviewers and the following data and other policy-related requests.

IMPORTANT - please attend to the following:

a) Please address the remaining comments from reviewer #1. The Academic Editor has kindly provided some additional guidance (see the foot of this email) which may prove helpful when deciding how to address the remaining issues.

b) Please address my Data Policy requests below; specifically, we need you to supply the numerical values underlying Figs 1ABCDEF, 2ABCDE, 3AC, 4ABCD, 5BCDE, 6ABCDE, S1ABCDEFGHIJKLMNOP, S2, S3, S4, S5, S6, S7, S8, S9, S10, S11, S12, S13, S14, S15, S16, S17, S18, S19, S20, S21, S22, S23ABCDE, either as a supplementary data file or as a permanent DOI’d deposition. This should include any treefiles. I note that you already have an associated Zenodo deposition, but this currently only seems to contain raw data and one piece of Python code. Please could you complete this deposition with the data and code needed to recreate the Figures?

c) Please cite the location of the data clearly in all relevant main and supplementary Figure legends, e.g. “The data underlying this Figure can be found in S1 Data” or “The data underlying this Figure can be found in https://zenodo.org/records/XXXXXXXX

d) Please make any custom code available, either as a supplementary file or as part of your data deposition.

e) Please include the URLs of your funders in the Financial Disclosure statement.

We expect to receive your revised manuscript within two weeks.

*Published Peer Review History*

*Press*

Sincerely,

Roli Roberts

Roland Roberts, PhD

Senior Editor

rroberts@plos.org

PLOS Biology

DATA POLICY:

Regardless of the method selected, please ensure that you provide the individual numerical values that underlie the summary data displayed in the following figure panels as they are essential for readers to assess your analysis and to reproduce it: Figs 1ABCDEF, 2ABCDE, 3AC, 4ABCD, 5BCDE, 6ABCDE, S1ABCDEFGHIJKLMNOP, S2, S3, S4, S5, S6, S7, S8, S9, S10, S11, S12, S13, S14, S15, S16, S17, S18, S19, S20, S21, S22, S23ABCDE. NOTE: the numerical data provided should include all replicates AND the way in which the plotted mean and errors were derived (it should not present only the mean/average values).

CODE POLICY

DATA NOT SHOWN?

REVIEWERS' COMMENTS:

Reviewer #1:

The updated manuscript is much improved. The authors have addressed many of my concerns. The restructured section on the kmer frequency analyses, in particular, is substantially improved on clarity and scientific veracity. The incorporation of kmer analyses on the experimental dataset is intelligent, cogent, and commendable.

Regarding the genome size estimates, I agree with the author's assessment that there are broad phylogenetic signals even if individual estimates are imperfect. I also agree that these signals are sufficient for the paper's analyses. However, as to not set a bad precedent for others to equate assembly size with genome size, I strongly recommend that the authors should include a clear statement in the manuscript indicating that the genome size estimates presented are genome assembly sizes which are imperfect and typically poor approximations (and usually underestimates) of the true genome size.

Regarding the result of 30-fold genome expansion, I may be missing something but this massive increase does not seem to be reflected in the color of the phylogenetic branches in Figures 2E or 3C. It's a bit hard to tell for the former due to obstructing graphical elements, but for the latter, the branch barely changes colour and remains relatively gray even though an increase this large should result in the branch becoming very red.

For figure 3B, the plot is now better labeled, but it is still not clear to me that the correct orthologous region was identified. Many of the RID1 locus genes in Cadophora have no homology to the "syntenic" region in Blumeria. Reciprocally, many of the Blumeria genes also have no homology to the supposed syntenic region in Cadophora. Based on these tracks, it can only be concluded that the RID1 locus in Cadophora is highly rearranged compared to Blumeria. Importantly, two genes upstream of Rid1 (green) also have no homology, so one interpretation could be that RID1 and the two upstream neighbors translocated into or out of this region. Can the orthologs of the two genes upstream of RID1 be found elsewhere in the genome?

Lines 257: The heading "Genes likely to impact RIP were lost in fungi" implies that all fungi lack these genes, when the authors mean there are recurrent lineage-specific losses of the genes.

The authors switch between different taxonomical hierarchies, sometimes using phylum, other times using class or genus. For the mycologists, this might all seem very natural, but this will be quite confusing for a more general audience.

In the abstract (lines 44-46) and discussion (lines 449-452) the authors imply a very close relationship between RIP and genome size. However, based on their results, this relationship seems to be relevant to ascomycota where there is clear RIP signatures, and perhaps leotiomycetes which has a branch with 30-fold increase in the absence of RIP. However, a glaring counter point to this pattern is the Saccharomycetes which has small to modest genome sizes but completely lack the RIP genes. The author acknowledges this later in the discussion (lines 499-502), pointing to trophism as a better predictor. Therefore the tone of the abstract must be reconsidered as to not give the false impression that RIP or TE regulating mechanisms drive genome size across the kingdom.

Reviewer #2:

The revision adequately addresses all my comments.

COMMENTS FROM THE ACADEMIC EDITOR:

On reviewer number 1:

1) I agree that there is something odd about the color scheme in Figure 3A. I think that perhaps the units are not the same as in Figure 2E. Figure 2E is in Mb and I think 3A might be proportional to the maximum genome size which explains why the basal branches are "cold" and the max value is 100 and not 1,000. This is minor but does need to be clarified.

2) I agree authors should look for the two upstream genes in Fig 3B and perhaps add them to the panel. But i think their overall interpretation does not differ from Reviewer #1: these regions are highly rearranged that potentially indicates relaxed purifying selection.

I looked over comments from reviewer number 4 and I feel they were reasonably addressed. The major comment regarding the Kmer frequencies was addressed nicely with the mutant analysis. Other comments were mostly overlapping with other reviewers to my eye.

---

## [Editor Report · Decision Letter 2]

23 Sep 2025

Dear Thomas,

Thank you for the submission of your revised Research Article "Phylogenomic signatures of repeat-induced point mutations across the fungal kingdom" for publication in PLOS Biology. On behalf of my colleagues and the Academic Editor, Erin Kelleher, I'm pleased to say that we can in principle accept your manuscript for publication, provided you address any remaining formatting and reporting issues. These will be detailed in an email you should receive within 2-3 business days from our colleagues in the journal operations team; no action is required from you until then. Please note that we will not be able to formally accept your manuscript and schedule it for publication until you have completed any requested changes.

Sincerely, 

Roli

Senior Editor

PLOS Biology

rroberts@plos.org